# Pericytes orchestrate a tumor-restraining microenvironment in glioblastoma

Sebastian Braun [1,6], Paulina Bolivar [1,6], Clara Oudenaarden[2], Jonas Sjölund [1], Matteo Bocci [1], Katja Harbst [3], Mehrnaz Safaee Talkhoncheh[1], Bengt Phung[3], Eugenia Cordero[4], Rebecca Rosberg [1], Elinn Johansson [1], Göran B. Jönsson [3], Alexander Pietras [1] & Kristian Pietras [1,5] ✉

Glioblastoma (GBM) is characterized by fast progression, infiltrative growth pattern, and a high relapse rate. A defining feature of GBM is the existence of spatially and functionally distinct cellular niches, where malignant cells engage in paracrine crosstalk with cell types comprising the tumor microenvironment. Here, we identify pericytes as the most active paracrine signaling hub within the tumor parenchyma. Their depletion through genetic engineering results in accelerated tumor progression and shortened survival. Mechanistic studies reveal that pericyte deficiency remodels the endothelium and impacts the immune cell landscape, exacerbating tumor cell invasion and immune suppression. Specifically, the pericyte-deprived endothelium recruits perivascular, tumor-associated macrophages polarized towards an immune-suppressive phenotype. The recruited macrophages express Hepatocyte Growth Factor, which reinforces activation of its receptor tyrosine kinase MET on GBM cells harboring a pronounced mesenchymal subtype driven by the key phenotypic regulator *Fosl1*. Indeed, orthotopic implantation of MET-expressing GBM cells corroborates their superior tumor-initiating and invasive capabilities. Thus, pericytes represent critical modulators of GBM development by orchestrating a tumor-suppressive microenvironment, highlighting the importance of their preservation in therapy.

Glioblastoma (GBM) is an aggressive high-grade brain tumor that, despite significant efforts over the last decades, remains resistant to current post-surgical therapeutic approaches[1,2]. The dismal prognosis of GBM is likely a result of diverse intrinsic features, including intra-tumoral heterogeneity, phenotypic plasticity, and a complex interplay between tumor cells and their microenvironment[3,4]. The latter is manifested as a complex tumor ecosystem, consisting of inter-connected but spatially and functionally distinct niches, such as the leading-edge niche, the hypoxic niche, and the perivascular niche

(PVN)[5]. Recently, it has become clear that glioma cells co-evolve with such niches and form specific molecular cell states that are adapted to the respective microenvironment[6]. Remarkably, due to a high degree of glioma cell intrinsic plasticity, gliomas can also react to and escape from environmental pressures such as lack of oxygen or ther-apeutically induced stress[7]. However, on a higher level of granularity, our understanding of tumor, stromal, and immune cell interactions is still inadequate, and the nature and sequence of the processes driving glioma cell adaptation remain elusive.

[1]Division of Translational Cancer Research, Department of Laboratory Medicine, Lund University Cancer Centre, Lund University, Lund, Sweden. [2]Biotech Research and Innovation Centre, University of Copenhagen, Copenhagen, Denmark. [3]Division of Oncology and Pathology, Department of Clinical Sciences, Lund, Lund University Cancer Centre, Lund University, Lund, Sweden. [4]Department of Clinical Sciences, Malmö, Lund University Diabetes Centre, Lund University, Malmö, Sweden. [5]SciLifeLab, Department of Laboratory Medicine, Lund University, Lund, Sweden. [6]These authors contributed equally: Sebastian Braun, Paulina Bolivar. ✉e-mail: kristian.pietras@med.lu.se

The GBM vasculature is characterized by microvascular proliferation and functional abnormality[8]. It represents a crucial factor for tumor development and dissemination: glioma cells are presumably guided by chemotactic factors secreted by the vasculature, and contact-dependent interactions; processes recapitulating the physiologic migration patterns of glial progenitor cells that allow for fast and diffuse permeation of the surrounding brain parenchyma along blood vessels[9,10]. Moreover, tumor-driving, stem-like glioma cells have been found to prosper along micro-vessels[11,12]. Given the character of the PVN as a crosspoint of tumor and stromal cells, and a crucial infrastructure for brain infiltration, deepened insights into how the aberrant GBM vasculature contributes to glioma progression are much needed and may support the design of more effective therapies.

Pericytes, mesenchymal smooth muscle-like cells, own a key position in the PVN due to their location: they are embedded in the basal lamina of microvessels and in contact with both endothelial and tumor cells[13]. Whereas pericytes have been extensively described to exert functions as regulators of vascular morphogenesis, angiogenic quiescence, and blood-brain barrier patency, consequently maintaining vessel homeostasis in the physiologic brain context[14,15], their role in brain tumors is still debated; pericytes have been linked to GBM progression[16], the induction of tumor immune-tolerance, shielding glioma cells from effective T-cell responses[17], and resistance to chemotherapy[18]. Furthermore, it has been suggested that a significant proportion of GBM pericytes are derived from glioma cells[19], and that pericytes should be used as drug targets to improve the treatment of brain tumors[20]. Emerging evidence, however, demonstrates heterogeneity among the group of mural cells, and that pericytes are not the only mesenchymal stromal cells to reside at glioma capillaries[21–23], demanding a deeper characterization of glioma developmental processes related to true pericytes.

Here, by analysis of single-cell transcriptomic data of human GBM, we unexpectedly identified pericytes as the most active paracrine signaling hub within the tumor parenchyma. This finding prompted us to study the effects of pericyte depletion in orthotopic and transgenic mouse models of high-grade glioma using single-cell and spatial transcriptomics, as well as multiplex immunohistochemistry and mechanistic studies. We present evidence that pericytes mitigate glioma development by orchestrating a tumor-suppressive microenvironment, challenging the concept of pericytes as merely prooncogenic cells, and underscoring the importance of pericyte preservation in future GBM therapies.

## Results

### Pericytes represent a signaling hub in human GBM
We commenced our investigations with the goal of characterizing the heterogeneity of glioma, stromal, and immune cells, and to better understand how the cells forming the tumor TME communicate and interact within different niches. Therefore, we leveraged a public dataset, comprising single-cell RNA-sequencing (scRNA-seq) of 201,986 tumor, immune, and stromal cells, derived from two low-grade and 16 high-grade human glioma samples[24]. In total, seven main cell clusters were annotated, based on well-known cell type markers, representing glioma cells, B- and T-cells, myeloid cells, endothelial cells, and oligodendrocytes (Figs. 1A and S1A). A pericyte cluster was defined based on the expression of established marker genes such as *S1PR3*, *ABCC9*, *CD248*, *RGS5*, *ANPEP*, *CSPG4*, *PDGFRβ*, *HIGD1B*, *NDUFA4L2*, *VTN* and *KCNJ8*[25] (Fig. 1B). To learn more about the interactions between the different tumor, immune and stromal cell compartments, we applied two algorithms, CellChat[26] and CellPhoneDB[27], that use different computational approaches to quantitatively characterize intercellular communication networks[26,28]. Intriguingly, both algorithms showed pericytes to be the cell type exerting the highest number of paracrine interactions, most notably with endothelial and glioma cells (Figs. 1C and S1B, C). Pericytes showed a clear dominance

in signaling towards other cell types, implying orchestration of distinct niches (Fig. 1D). Despite the low relative abundance of pericytes within the GBM microenvironment, spatial expression analysis of the mural cell marker genes *PDGFRβ* and *ACTA2* in a human spatial transcriptomics GBM dataset (Fig. S1D) demonstrated the strategic positioning of pericytes in close proximity to blood vessels with easy access to the malignant parenchyma. The vascular abundance of pericytes was further corroborated by patient samples immunostained for PDGFRβ and ACTA2 (images available from v24.0.proteinatlas.org[29,30]) (Fig. S1E, see "Data availability" statement). Together, these data demonstrate a putative crosstalk between pericytes, glioma cells, and the TME in glioma, the functional nature of which we set out to determine.

### Pericyte deprivation leads to an accelerated tumor progression and premature death in two glioma mouse models
To functionally assess the role of pericytes in glioma development and progression, we stereotactically injected *GBM* cells into the subventricular zone (SVZ) of wild-type mice or of mice carrying a homozygous *Pdgfb* allele that is missing the PDGF-B heparan sulfate proteoglycan binding motif sequence (*Pdgfb^ret/ret* mice)[31]. In these mice, PDGF receptor-β (PDGFRB)-expressing pericytes, which under normal circumstances are recruited to nascent blood vessels through PDGF-BB secretion by endothelial tip cells, are not properly investing the vessel wall due to the increased diffusion of the PDGF-BB ligand in *Pdgfb^ret/ret* mice (Fig. 2A). Hence, a high degree of pericyte loss in the vasculature (70–80%) has been validated in several reports[14,22,31–33]. Interestingly, following stereotactic engraftment of *p53^−/−*, *H-Ras* overexpressing neurospheres[34] into the SVZ, *Pdgfb^ret/ret* mice exhibited a significantly shorter survival time compared to *Pdgfb^ret/+* and wild-type animals (Fig. 2B). Upon necropsy, the brains of pericyte-poor mice presented with overtly hemorrhagic lesions containing necrotic areas (Fig. 2C). In accordance with the literature, we observed an estimated 70–80% reduction of pericyte coverage of tumor vessels upon implantation of syngeneic GL261 cells[35] in *Pdgfb^ret/ret* mice, compared to control mice (Fig. 2D, E). We validated our findings by utilizing RCAS virus-mediated induction of oncogenic *PDGFB* expression and knockdown of *Tp53* in SVZ cells of mice expressing the tv-a receptor under the *Nestin* promoter (Ntv-a), resulting in tumors that resemble human oligodendroglioma and display a high-grade histology[36,37]. Accordingly, we crossed *Pdgfb^ret/ret* mice with Ntv-a mice and induced tumor formation through intracranial injection of virus-producing DF-1 cells. Confirming our previous results, we observed both a dramatically reduced pericyte vessel coverage and a significantly shorter latency until the *Pdgfb^ret/ret* mice succumbed to the progressing brain tumors (Fig. 2B, E, F). The similar results in two different GBM models driven by distinct oncogenic pathways demonstrate that the accelerated tumor progression in pericyte-deprived mice is independent of the genetic engineering of the *Pdgfb^ret/ret* mice. To better understand the biological processes leading to an aggravated tumor progression upon pericyte depletion, we characterized the tumor development more profoundly. To measure hypoxia in the glioma tissue of our mouse model, we quantified HIF2α expression, together with an OLIG2 analysis that served as an indicator of tumor boundaries and infiltrative areas, based on immunostaining, as well as performed a flow cytometry-based analysis of pimonidazole (PIM) adducts in PDGFB-induced brain tumors from cohorts of *Pdgfb^ret/ret* and *Pdgfb^ret/+* mice. Regardless of methodology, we did not detect a significant difference in hypoxia in gliomas of pericyte-deprived mice (Figs. S2A, B). In contrast, a proliferation analysis, based on the expression of Ki67, demonstrated a greater accumulation of Ki67+ cells in PDGFB-induced tumors from *Pdgfb^ret/ret* mice compared to *Pdgfb^ret/+* littermates (Fig. 2G, H). To learn more about the tumor growth dynamics, we next transplanted luciferase-expressing *p53^−/−*, *HRAS* over-expressing glioma cells intracranially into *Pdgfb^ret/ret* and *Pdgfb^ret/+* animals and monitored the

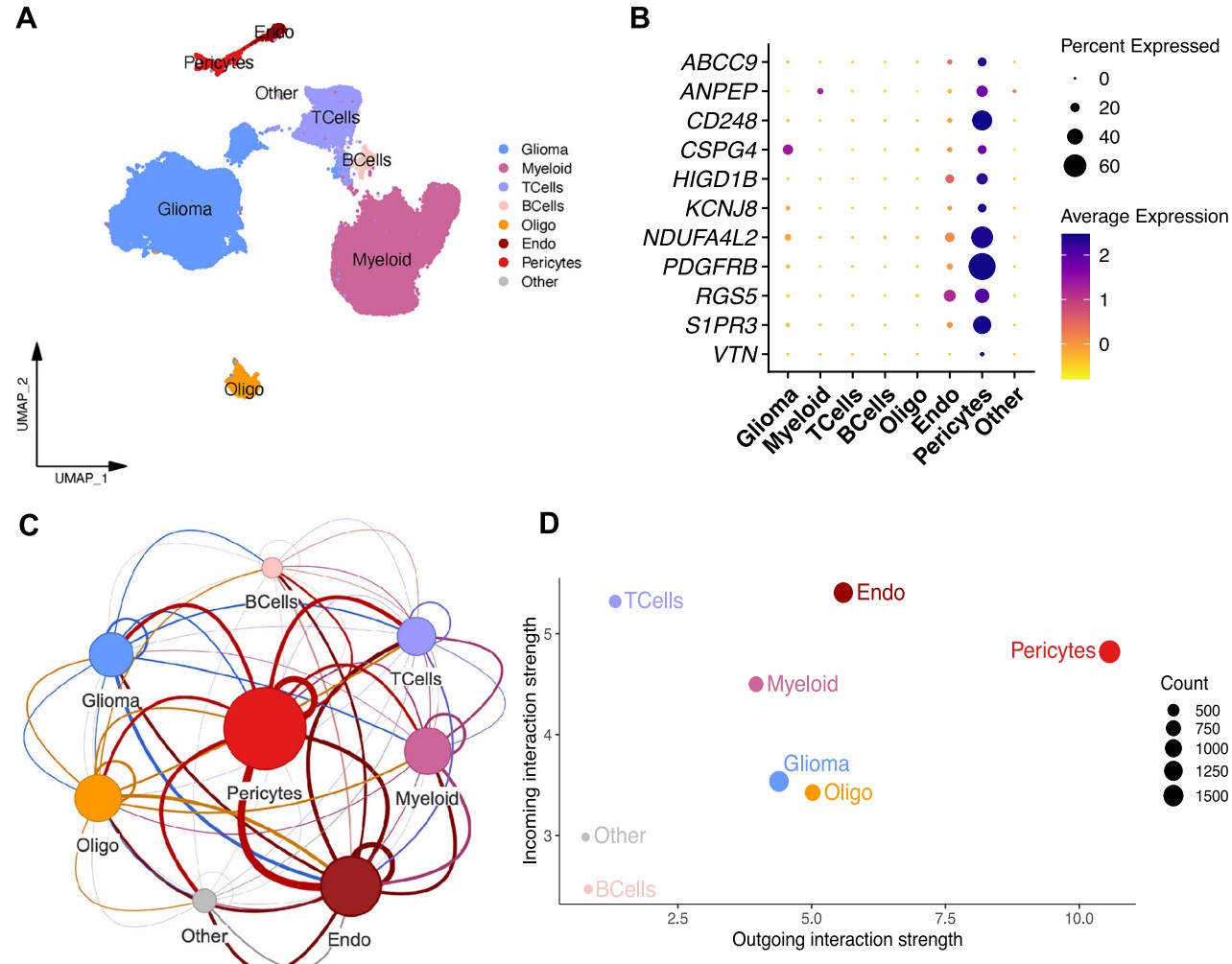

**Fig. 1 | Pericytes form a pivot of intercellular communication in human high-grade glioma. A** UMAP plot of human glioma cells analyzed with scRNA-seq. Color-coding corresponds to major cell lineages. **B** Average expression and percentage of cells expressing selected prototypical pericyte marker genes across the cell groups. **C** A network of human glioma cell clusters, showing the overall cellular communication (autocrine and paracrine), as assessed by CellChat. Node sizes represent the total number of interactions weighted by the communication strength, while edges show the weighted number of paracrine interactions between two clusters. Autocrine interactions are represented by edges that fall back on the same cluster. **D** Scatterplot showing the total incoming or outgoing interaction strength associated with each cell population. Dot sizes represent the number of inferred connections (both outgoing and incoming). Data and annotations from Abdelfattah et al.[24].

subsequent tumor growth by measuring the total luciferase biolumi-nescence. In a direct comparison, we found *Pdgfb*[ret/ret] mice to exhibit a higher luciferase activity than the control animals at all time points (Fig. 2I). To better estimate differences in the growth kinetics of *Pdgfb*[ret/ret] and *Pdgfb*[ret/+] mice, we fit the data to a non-linear model and observed a higher glioma growth rate in *Pdgfb*[ret/ret] mice over time (Figs. 2J and S3), indicating a growth advantage over *Pdgfb*[ret/+] gliomas, in keeping with their higher proliferative rate and faster induction of end-stage disease.

In conclusion, our experimental data reveal that loss of pericytes sustained the engraftment and progression of highly proliferative brain tumors.

**Pericyte deprivation remodels the glioma perivascular niche and impacts the immune cell landscape**

Based on our observations that pericytes are strong paracrine signal senders in glioma and instruct both tumor and stromal cells, and that their depletion exacerbates tumor progression, we aimed to better understand how pericytes influence the different glioma compart-ments and their interactions. To that end, we performed scRNA-seq of

dissociated tumors from four *Pdgfb*[ret/ret] and seven *Pdgfb*[ret/+] mice, car-rying *PDGFB* over-expressing, *p53* knockdown, RCAS virus-induced glioma tissues. Unsupervised, graph-based clustering recognized 21 clusters in the integrated dataset (Fig. 3A). To annotate clusters, we compared significantly over-expressed genes of each cluster to literature-based cell markers[38–45], using the top 50 differentially expressed genes (DEGs) of each cluster[46] (Fig. 3B). Tumor cells were identified based on the expression of an RCAS virus-unique gene sequence (Fig. 3C). We categorized the clusters into four main groups, namely immune clusters annotated as microglia, macrophages and NK/T/NKT cells (I); a tumor cell-enriched group representing cells expressing marker genes for the astrocytic, oligodendrocytic or neu-ronal lineages, and tumor cells of hypoxic and mesenchymal character (II); dividing cells (mostly malignant) (III), and vascular cells (IV)[36,38]. To test whether our mouse model reflects the malignant cell states that are commonly observed in human patients, we used gene signatures provided by Neftel et al.[44] and Couturier et al.[38]. In short, for each gene signature, we calculated the mean *Z*-score for the individual cells in our dataset. We did the same for our own cluster signatures. Then, we computed a pairwise correlation matrix between all signature scores

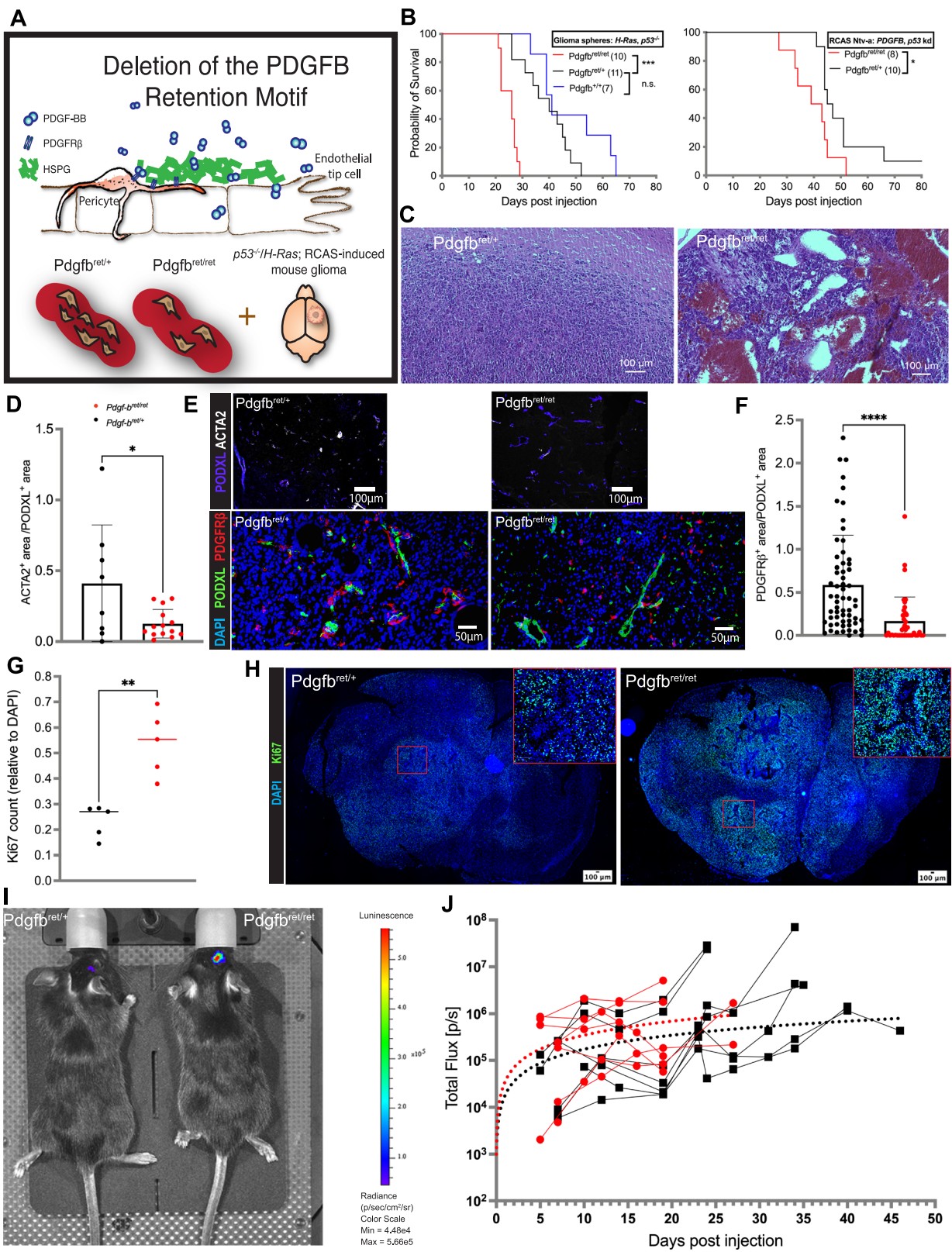

and applied hierarchical clustering to group signatures, based on how similarly they were expressed across our dataset (both *Pdgfb^{ret/ret}* and *Pdgfb^{ret/+}* tumor samples, Fig. 3D). The clustering of the tumor and dividing cell signatures from Neftel et al.[44] with the signatures from all our tumor clusters demonstrated the presence of intra-tumoral heterogeneity and proliferating cells (Figs. 3D, E and S4A). Notably, the MES2/MES-like signature grouped with our clusters MES I and II, and

the AC-like and "non-neural" signatures from Neftel et al.[44] and Couturier et al.[38] associated with our cluster AC (Figs. 3D and S4A). Except for a tendency towards a more pronounced NPC-like tumor compartment, we did not observe any major deviation regarding the "Neftel" cell states in tumors from *Pdgfb^{ret/ret}* as compared to *Pdgfb^{ret/+}* mice (Fig. S4B, C), reflecting the resilience of the dominant OPC/NPC state in this mouse model, and indicating that the phenotypical

**Fig. 2 | Alterations in glioma development upon pericyte reduction occur in different mouse models. A** Schematic of tumor cell engraftment or virus-mediated glioma generation in *Pdgfb*$^{ret/ret}$ mice. **B** Kaplan–Meier curves, showing symptom-free survival of *Pdgfb*$^{ret/ret}$, *Pdgfb*$^{ret/+}$ and *Pdgfb*$^{+/+}$ mice transplanted with *pS3*$^{-/-}$, *H-Ras* over-expressing tumor cells (left panel) and RCAS-PDGFB/shp53 virus-producing DF-1 cells (right panel). The number of mice analyzed for each model is indicated in parentheses. Log-ranked test, $p > 0.0001$, chi-square = 27.54, df = 2 (left panel), $p = 0.0408$, chi-square = 4.185, df = 1 (rigth panel). **C** Representative H&E fields from *pS3*$^{-/-}$/*H-Ras* tumor cell-induced gliomas in *Pdgfb*$^{ret/ret}$ and *Pdgfb*$^{ret/+}$ mice. Pericyte vessel coverage in GL261 tumor cell (**D**, upper panels in **E**) and PDGFB-induced (**F**, lower panels in **E**) gliomas in *Pdgfb*$^{ret/ret}$ and *Pdgfb*$^{ret/+}$ mice, presented as ACTA2$^+$/PODXL$^+$ and PDGFRβ$^+$/PODXL$^+$ ratios, derived from the area quantification analysis of immunostained glioma sections (**E**). Three *Pdgfb*$^{ret/+}$ and four *Pdgfb*$^{ret/ret}$ mice were used for the GL261 analysis, and five *Pdgfb*$^{ret/+}$ and three *Pdgfb*$^{ret/ret}$ mice were used for the PDGFB-induction-based analysis. Each dot represents a tumor section taken from a distinct tumor position. Mean (open bars) and SD are shown. Two-sided *t*-test, $p = 0.021$, $t = 2.496$, df = 20 (**D**), $p < 0.0001$, $t = 4.315$, df = 99 (**E**). Proliferation analysis of *Pdgfb*$^{ret/ret}$ ($n = 5$) and *Pdgfb*$^{ret/+}$ ($n = 5$) PDGFB-induced gliomas, presented as Ki67$^+$ cells relative to DAPI (**G**), quantified from immunostainings (**H**). Median is indicated. Two-sided *t*-test, $p = 0.0014$, $t = 4.796$, df = 8 (**G**). Boxes denote the enlarged regions. **I**, **J** IVIS imaging of tumor-bearing mice from day 5 to 47 after transplantation of luciferase-expressing, *pS3*$^{-/-}$/*H-Ras* tumor cells. **I** Representative IVIS scan of a *Pdgfb*$^{ret/ret}$ and a *Pdgfb*$^{ret/+}$ mouse, 15 days post-transplantation. **J** Total photon flux, measured continuously from 8 *Pdgfb*$^{ret/ret}$ and 7 *Pdgfb*$^{ret/+}$ mice, was fitted to a non-linear model (dotted curves), representing an estimate of the tumor growth kinetics. Each dot represents the photon flux measurement of one mouse at a certain time point, each line represents a mouse included in the analysis. For plotted data, *Pdgfb*$^{ret/+}$ is depicted in black and *Pdgfb*$^{ret/ret}$ in red. *$p < 0.05$, **$p < 0.01$, ***$p < 0.001$, ****$p < 0.0001$, ns not significant. Source data are provided as a Source data file.

aberrance that we observed in glioma development upon pericyte removal was due to alterations in non-cell autonomous aspects of glioma biology.

Based on our findings that pericytes are extensively communicating with both tumor and non-tumor cells, we anticipated an altered glioma microenvironment to contribute to the premature death of pericyte-deprived mice. Therefore, we focused our attention on the tumor stroma, commencing with the PVN. Pericytes and endothelial cells could be identified in our dataset based on the expression of marker genes such as *Kcnj8*, *Pdgfrb* and *Higd1b*, as well as *Cldn5*, *Tie1*, and *Dll4*, respectively (Fig. 3B). Indeed, in keeping with our findings in human tumors, when we analyzed cell-cell communication networks we found pericytes to exert the most paracrine interactions in both *Pdgfb*$^{ret/ret}$ and *Pdgfb*$^{ret/+}$ tumors (Fig. S5A, B).

We next wanted to understand if structural changes occurred upon the deprivation of pericytes. To further validate the *Pdgfb*$^{ret/ret}$ mouse model, we quantified the number of cells comprising the PC and EC clusters and found, as anticipated, the amount of pericytes to be reduced by 51% in *Pdgfb*$^{ret/ret}$ mice. In sharp contrast, the EC population was increased by 40% (Fig. 3A), in line with previous findings from investigations on brain angiogenesis[47]. Moreover, measurements based on the labeling of the endothelium with antibodies directed against Podocalyxin and PECAM1 provided evidence of structural differences between the *Pdgfb*$^{ret/ret}$ and *Pdgfb*$^{ret/+}$ glioma vasculature, as we found vessel area, vessel length, and number of junctions to be significantly higher in pericyte-poor tumor tissue (Figs. 4A, B and S5C). Finally, we performed a functional analysis of the intactness of the blood-brain barrier by perfusion with labeled dextrans (70 kDa) that indicated that vascular fluids leaked into the perivascular space (Fig. 4C), revealing a significant loss of vascular integrity upon pericyte reduction. The findings of a corrupted endothelium due to pericyte deprivation raised the question of whether a structurally altered PVN would have consequences for glioma progression. Since the concept of tumor cell migration along endothelial routes in a vessel co-opting manner is well established[10,48,49], we next aimed at quantifying this process in *Pdgfb*$^{ret/ret}$ and *Pdgfb*$^{ret/+}$ gliomas. Based on the observation that the vast majority of glioma cells in our mouse model are of OPC/NPC character, expressing *Olig2*, and that these cells tend to invade the brain by single-cell vessel-cooption[50], we performed an immunostaining analysis and quantified the number of OLIG2$^+$ cells related to endothelial cells at the invasive rim of mouse tumor-brain tissue sections. We observed two different invasion modes of OLIG2$^+$ tumor cells, namely the diffuse spread of individual cells into the surrounding physiological brain parenchyma and vessel co-option of single cells. Interestingly, we did not detect differences regarding the total number of invading OLIG2$^+$ tumor cells in the invasive zone but found a significantly higher proportion of vessel co-opting tumor cells in *Pdgfb*$^{ret/ret}$ animals (Figs. 4D, E and S5D), indicative of a more permissive PVN in the absence of pericytes.

Next, we set out to investigate putative alterations of the immune cell landscape in pericyte-poor tumors. FACS-based quantification of different immune cell compartments showed a significant increase in CD45$^+$ cells and a strong trend toward a decrease in dendritic cells for *Pdgfb*$^{ret/ret}$ tumors (Fig. S5E). Since an accumulation of myeloid cells with tumor-supportive features has been shown for experimental pericyte-poor tumors[51], we focused on the prevalence of immune-suppressive, polarized macrophages that are typically characterized by the expression -among others- of *Cd206*, *Cd204*, *Arg1*, and *Lgals3*[52]. In tissue sections of tumors from both *Pdgfb*$^{ret/ret}$ and *Pdgfb*$^{ret/+}$ mice, we found CD206$^+$ macrophages to be exclusively associated with blood vessels (Fig. 4F), indicating a preference for, and/or an induction of a polarized phenotype by the perivascular space. Using flow cytometry, we quantified two populations of alternatively activated myeloid cells that have been described as immune-suppressive, defined by positivity for CD206, CD11b, and F4/80 (alternatively activated macrophages), as well as the marker expression combination CD11b$^+$/LY6C$^+$/LY6G$^-$ (myeloid-derived suppressor cells, MDSC), and observed a significantly higher cell number in *Pdgfb*$^{ret/ret}$ tumors for both cell populations (Fig. 4G). Intriguingly, we also noted a shift towards a more extreme polarization for macrophages in pericyte-deprived tumors, as these cells exhibited a higher expression of genes associated with alternative activation, such as *Retnla*, *Ccl24*, *Mrc1* (CD206), *Arg1*, and *Klf4* (Figs. 4H and S6A). Moreover, a gene ontology (GO) over-representation analysis with Metascape[46] revealed alterations in inflammatory response pathways in *Pdgfb*$^{ret/ret}$ tumors for the microglia and macrophage compartments, and alterations in leukocyte activation and cytokine production exclusively for macrophages (Fig. S7A). Whereas an increased accumulation of myeloid cells could, at least partially, be explained by a more permeable vasculature, this structural alteration is unlikely to impact the character of macrophages. Therefore, we scrutinized the signaling profile of the endothelium in our scRNA-seq data with CellChat[26]. Of note, three signaling axes known to induce macrophage polarization—CD200, CSF, and GAS6[53–55]—were exclusively active in *Pdgfb*$^{ret/ret}$ endothelial cells that signaled towards macrophages (Figs. 4I and S8A), and possibly instructed them toward an immune-suppressive phenotype. Interestingly, when we added one of the ligands that were over-expressed in the pericyte-deprived PVN, GAS6, to primary cultures of mouse bone marrow-derived macrophages, we found GAS6 to reinforce IL4-induced alternative activation, indicating that GAS6 signaling may contribute to further polarization of macrophages that have already been primed in a tumor-supportive way (Fig. S8B).

Lastly, we investigated whether pericytes might directly exert a dampening, anti-polarizing effect on myeloid cells. To this end, we applied supernatant collected from mouse brain pericyte and endothelial cell cultures to proliferating macrophages and included IFNγ-"non-polarized" and IL4-"super-polarized" controls. Pericyte supernatant, in contrast to endothelial cell supernatant, dampened the IL4-

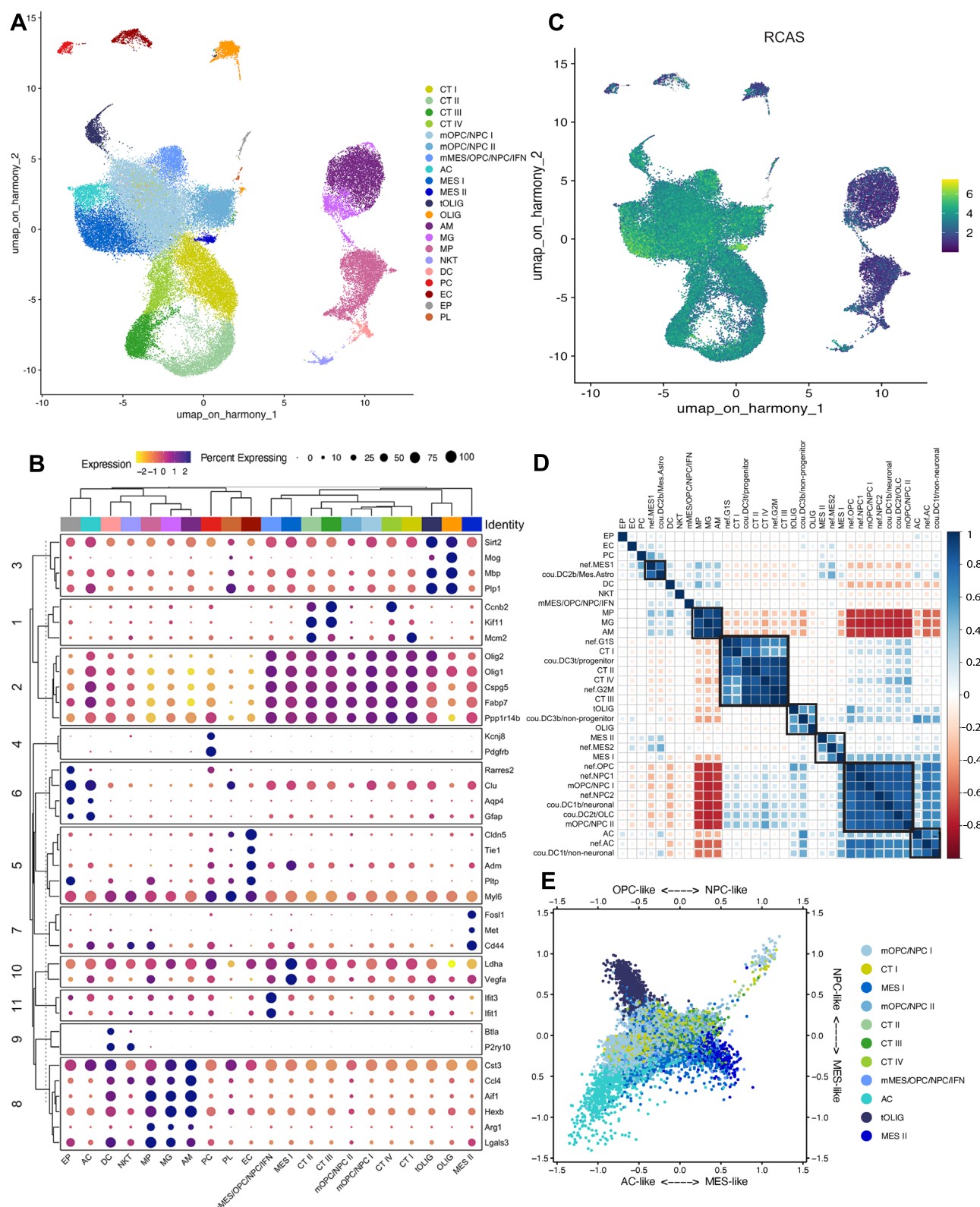

related "super-polarization" effect, indicating a putative direct paracrine, polarization-suppressing effect of pericytes (Fig. 4J) that deserves further corroboration.

In summary, our results show that the deprivation of pericytes leads to remodeling of the PVN characterized by dysregulated vessel growth, reduced patency of the blood-brain barrier, increased tumor cell vessel co-option, altered endothelial cell signaling, and an increased abundance of polarized immune-suppressive macrophages.

## A hypoxia-associated subset of GBM cells harbors a distinct mesenchymal state

Since previous work has shown that the MES-like GBM cell state is not only correlated to the abundance of macrophages, but directly driven by alternatively activated myeloid cells[45,56-58], we scrutinized the MES I and II cells of our scRNA-seq dataset more thoroughly. Among the 50 top DEGs in the MES I cluster, we found many to be related to hypoxia induction and regulation, such as *Vegfa*, *Ldha*, and *Pgk1*

**Fig. 3 | Molecular characteristics of glioma tumor and stromal cells derived from a pericyte-deprived mouse model. A** UMAP plot of 16,086 *Pdgfb*[ret/ret] and 42,838 *Pdgfb*[ret/+] cells that were isolated from a total of 4 and 7 tumors, respectively, and analyzed with scRNA-seq. The UMAP clusters are colored based on cell identity as cycling tumor cells I-IV (CT I-IV), mixed OPC-like/NPC-like tumor cells I-II (mOPC/NPC I-II), mixed mesenchymal/OPC-like/NPC-like/IFN tumor cells (mMES/OPC/NPC/IFN), astrocytes/AC-like tumor cells (AC), mesenchymal tumor cells I-II (MES I-II), tumor oligodendrocytes (tOLIG), oligodendrocytes (OLIG), activated microglia (AM), microglia (MG), macrophages (MP), NK/T/NKT cells (NKT), dendritic cells (DC), perivascular cells/pericytes (PC), endothelial cells (EC), ependymal cells (EP) and platelets (PL). **B** Clustered dot plot showing the scaled average expression of selected top differentially expressed genes across all clusters. **C** Feature plot showing the log-normalized expression of the RCAS-unique vector sequence across all clusters. **D** Clustered correlation matrix showing pairwise Pearson's correlation coefficients of the mean *Z*-score per cell estimated for different cell identity signatures from this study, Neftel et al. (nef), and Couturier et al. (cou). Black boxes highlight selected clustered signatures. **E** Butterfly plot of molecular subtype signature scores defined by Neftel et al. The quadrants correspond to the four subtypes: mesenchymal-like (MES-like), neural-progenitor-like (NPC-like), astrocyte-like (AC-like), and oligodendrocyte-progenitor-like (OPC-like). The position of each cell reflects its relative signature score across both axes. Colors represent the different clusters, as in (**A**). Source data are provided as a Source data file.

(Figs. 5A and S9A). Whereas the MES II cluster again identified hypoxia and stress-related genes like *Hmox1*, *Tnc*, *Gdf15*, and *Hspa9*, MES II marker genes also included *Fosl1*, *Met*, *Ccn4*, *Grb10*, and *Sema3c*, genes that have previously been associated with stemness, tumor cell plasticity and invasion, the regulation of EMT-like processes, and generally an increased tumor aggressiveness[59–63]. Moreover, a transcription factor analysis with DoRothEA[64–66] showed a similar activity pattern between MES I and MES II cells but indicated a much higher transcription factor activity for MES II cells (Fig. S9B). Interestingly, several MES II marker genes, such as *Ccn4*, *Tnc*, and *Grb10*, were co-expressed by pericytes, cells of mesenchymal character, sometimes also considered as mesenchymal stem cells[13]. Thus, whereas the expression profiles of both MES I and II cells appeared to be related to hypoxia, only the profile of MES II cells represented a panel of distinct mesenchymal features. Notably, the overall percentage of tumor cells decreased, whereas we observed the opposite trend for MES II cells when comparing *Pdgfb*[ret/+] versus *Pdgfb*[ret/ret] gliomas (34% increase in *Pdgfb*[ret/ret] mice; Figs. 3A and S4C), indicating that the loss of pericytes favors the expansion of this extreme mesenchymal glioma cell state. This notion was further supported by cell-cell communication analysis that demonstrated an increased signaling activity of pathways related to mesenchymal transformation, such as Wnt, Notch, and Gas6/Axl[67–69] in MES II cells of *Pdgfb*[ret/ret] samples (Figs. S8A, C and S9C).

To interrogate if MES II cells were more prevalent in certain tumor biomes[70], we explored our glioma samples in a spatial context using array-based, spatial transcriptomics RNA-sequencing (stRNA-seq) technology (10x Genomics Visium), and simultaneously visualized blood vessels, hypoxic regions, and proliferative tissue areas by immunostaining for Podocalyxin (PODXL), PIM adducts, and Ki67, respectively (Figs. 5B and S9D). To spatially characterize the tumor sections, we integrated the data of four *Pdgfb*[ret/ret] and four *Pdgfb*[ret/+] tumor sections and overlaid the transcriptomics data with the immunostaining images. The expression profile of each Visium spot is a mixture of the transcriptome of several cells that are likely of different cell types. Therefore, these clusters represent niches or tumor zones, rather than cell types. To annotate these clusters, we considered IF stainings, DEG, and estimated scRNA-seq cluster *Z*-scores. We annotated 14 spatial clusters of predominantly tumor or physiologic phenotype (Figs. 5C, S9E and S10A). Whereas we found cluster st_Hyp to overlap with hypoxic, PIM⁺ areas, and to feature a generally hypoxic tumor cell expression signature, these zones were surrounded by Ki67⁺ st_Prol spots of a dominant proliferation profile that highly correlated with our scRNA-seq proliferation signatures of clusters CT I-IV (Figs. 5C and S11A, B). Interestingly, st_Hyp-dominated tissue areas showed the strongest expression of the Neftel et al.[44] MES2 signature, the scRNAseq cluster MES II signature, and *Fosl1* (Figs. 5D, E and S11A, B). Moreover, macrophage polarization marker *Msr1* (CD204) appeared to be mostly expressed in hypoxic zones (Fig. S12A). Of note, we observed st_MF capture spots that had a high expression of genes typical for myofibroblasts (*Col1a1*, *Col1a2*, *Lum*, *Dcn*), interspersed among st_Prol and st_Hyp capture spots (Figs. 5C, S9E and S10A). Furthermore, we found st_Tumor to be spatially separated from

st_Hyp, st_Prol, and st_MF and to feature a mixed AC, mOPC/NPC I-like expression profile (Figs. 5C and S11A, B). The remaining spatial clusters were either annotated based on their expression of genes that are functionally related to neuronal processes and considered as glioma cell invasion zones (st_Phys1 and st_Phys2), or their spatial occurrence at meningeal zones and brain cavities lined by epithelial cells (st_Men and st_EP), or could not be specified (st_6-11) (Figs. 5C and S11B).

Clustering of the integrated spatial clusters based on selected DEGs corroborated our annotations, showing that the Visium capture spots were dominated by either a glioma/hypoxia/myofibroblast or a physiologic neural/meningeal expression profile (Fig. S10A). Moreover, to provide further support for the spatial cluster annotation and to characterize our samples on a deeper level, we estimated the cell type composition of spots by applying the Robust Cell Type Decomposition (RCTD) method[71]. RCTD infers the most likely proportion of different cell types in every Visium spot by using a scRNA-seq dataset as a reference. Here, we used the expression profiles from our 21 scRNA-seq clusters as reference and found, in almost perfect congruence with the mean *Z*-score estimations, RCTD to show a zonation of the tumor samples into a MES, an OPC/NPC, a proliferative, and a physiologic invasive zone, with the MES and the OPC/NPC sector being mutually exclusive (Fig. S11A).

Notably, the MES II signature *Z*-scores and the inferred proportion of MES II cells by RCTD for the st_Hyp clusters of the spatial dataset indicated an increase in MES II cells in *Pdgfb*[ret/ret] as compared to *Pdgfb*[ret/+] samples (Fig. S12B), which is in line with our observations from the scRNA-seq dataset. To determine the prevalence of MES II cells, we manually annotated the core hypoxic zones, as well as the peri-hypoxic zones of the spatial glioma sections, and visualized the distribution of MES II signature mean *Z*-scores and RCTD proportions across spots (Fig. 5F, G, H). We observed the highest expression of MES II signature *Z*-scores and the highest prevalence of MES II cells in the hypoxic core regions, and a gradual decrease of the expression intensity and prevalence towards the hypoxic periphery, while mOPC/NPC I cells exhibited a reverse distribution pattern.

Whereas this result confirmed hypoxic gene expression as a central element of the MES II cells, we wanted to find out more about the MES II signature genes that were functionally related to other mesenchymal glioma processes. To this end, we first asked whether there are differences in the spatial expression distribution of the scRNA-seq signatures of MES I and MES II in our stRNA-seq data (Figs. 6A and S11A). Of note, whereas MES I expression was restricted to hypoxic zones, MES II expression also appeared at the tumor invasive zone of the tissue section where tumor cells infiltrate physiologic areas of the brain parenchyma; a pattern that could be observed in most *Pdgfb*[ret/+] samples but was even more conspicuous in GBM from *Pdgfb*[ret/ret] mice (Fig. 6A). Additionally, the overall expression of the MES II signature was more pronounced in the absence of pericytes, in keeping with the higher abundance of these cells observed in the scRNA-seq analysis (Fig. 6A). These findings indicated alternative functions of MES II cells, exceeding a mere hypoxia-induced gene expression program. Interestingly, we found the MES II top marker

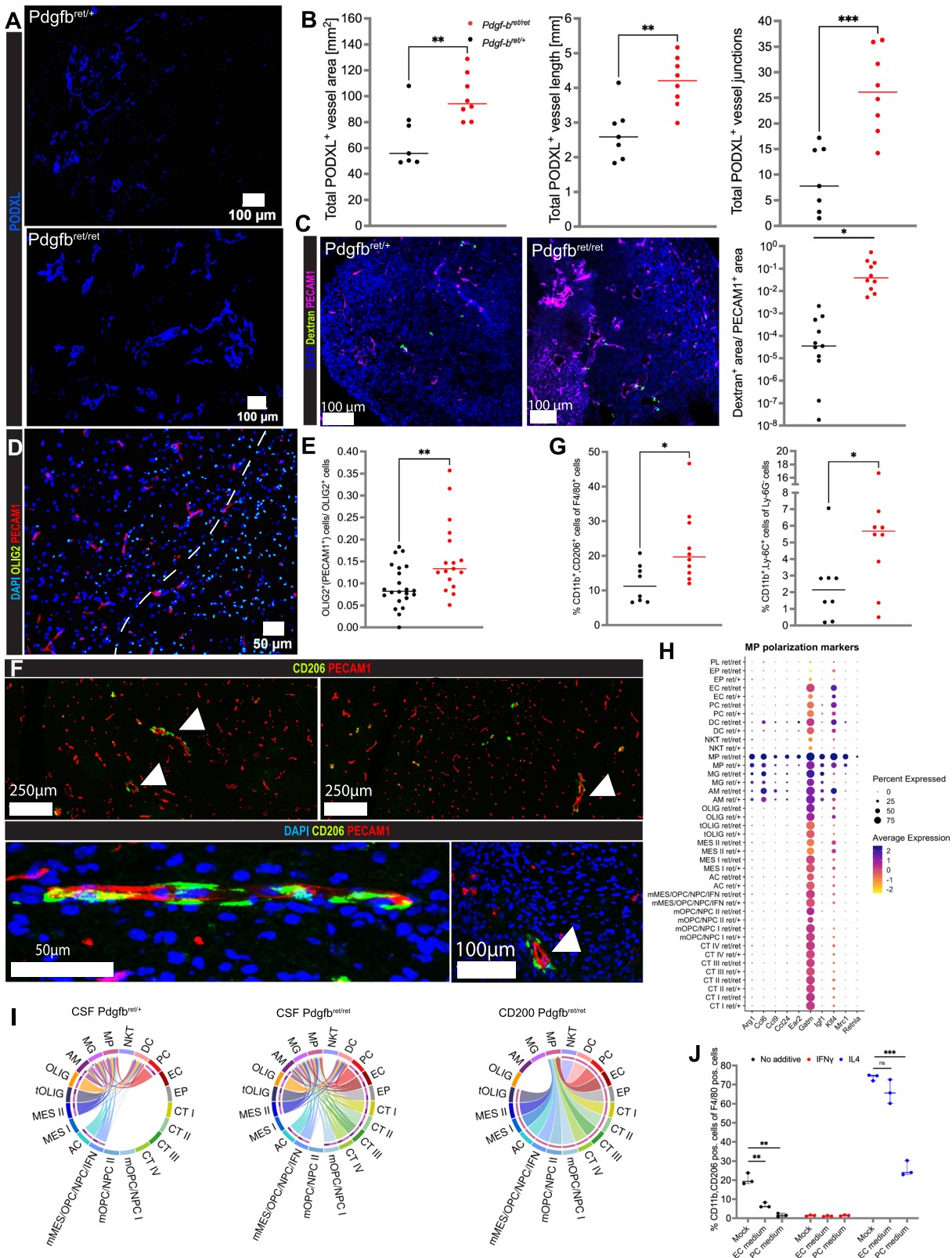

*Fosl1* to be expressed both in hypoxic and tumor peripheral zones. *Fosl1* is an AP-1 transcription factor, and its expression is correlated to tumor progression and unfavorable prognosis in various epithelial tumor types[72–75]. It has been found previously to be induced by hypoxia and represents a master regulator of stemness, EMT, and invasion[59,76,77]. Based on this observation, we next sought to explore if a subpopulation of MES II cells developed invasive capabilities and

migrated away from the st_Hyp/PIM+ zones, and toward the tumor periphery, linking absence of pericytes to increased invasiveness in *Pdgfb*ret/ret gliomas. Therefore, we first curated two MES II sub-signatures comprising genes associated with either hypoxia and stemness (HSS; Table S1) or invasiveness (ISS; Table S1). To see how unique these sub-signatures were, we checked their expression across the mouse glioma datasets (Fig. 6B, C). Whereas we found HSS to be

**Fig. 4 | Pericyte reduction causes remodeling of the perivascular and immune glioma microenvironment. A** Immunostainings of PODXL in *Pdgfb*^ret/ret^ and *Pdgfb*^ret/+^ *pS3*^−/−^/*H-Ras* tumor cell-induced glioma sections. **B** Assessment of glioma tissue parameters, comparing *Pdgfb*^ret/ret^ (*n* = 7) and *Pdgfb*^ret/+^ (*n* = 7) tissue sections of *pS3*^−/−^/*H-Ras* gliomas based on PODXL expression. Each dot represents the average of several FOVs in one tumor, taken at different positions. Area *p* = 0.0091, *t* = 3.064, df = 13. Length, *p* = 0.0025, *t* = 3.738, df = 13. Junction *p* = 0.0006, *t* = 4.511, df = 13. **C** Fluorescence images (left) and quantification (right) of a vessel functionality dextran (70 kDa) analysis comparing *pS3*^−/−^/*H-Ras Pdgfb*^ret/ret^ and *Pdgfb*^ret/+^ tumors. Dextran-positive areas were quantified and related to the PECAM1^+^ area. Each dot represents a tumor section, *p* = 0.0212, *t* = 2.370, df = 14. **D**, **E** Quantification of tumor cell infiltration at the invasive rim (IR) in PDGFB-induced *Pdgfb*^ret/ret^ and *Pdgfb*^ret/+^ tumors, based on PECAM1 and OLIG2 stainings. IR was defined as the FOV portion adjacent to the tumor core (white dotted line) (**D**). Vessel co-opting tumor cells were defined as OLIG2^+^ cells in contact with PECAM^+^ cells and normalized to total OLIG2^+^ cells. Each dot represents a FOV. *Pdgfb*^ret/ret^ (*n* = 17) and *Pdgfb*^ret/+^ (*n* = 22), *p* = 0.0038, *t* = 3.085, df = 37 (**E**). **F** PECAM1^+^ cell-associated CD206^+^ macrophages (*Pdgfb*^ret/ret^). Arrows indicate CD206^+^ cells adjacent to PECAM1^+^ cells. **G** Quantification of CD206^+^/CD11B^+^/F4/80^+^ macrophages (left) and CD11B^+^/LY6C^+^/LY6G^−^ myeloid-derived suppressor cells (MDSC, right), comparing PDGFB-induced *Pdgfb*^ret/ret^ (*n* = 10) and *Pdgfb*^ret/+^ (*n* = 8) tumors. Left panel *p* = 0.0212, *t* = 2.555, df = 16. Right panel *p* = 0.0447, *t* = 2.178, df = 16. **H** Scaled average expression of macrophage polarization markers in the scRNA-seq dataset, comparing *Pdgfb*^ret/ret^ and *Pdgfb*^ret/+^ cells. **I** Chord diagrams of pathways that exhibit different signaling patterns between *Pdgfb*^ret/ret^ and *Pdgfb*^ret/+^ tumors. TEdge weights are proportional to the estimated interaction strength. Left and middle panel: CSF pathway (active in both *Pdgfb*^ret/ret^ and *Pdgfb*^ret/+^ tumors); right panel: CD200 pathway (active exclusively in *Pdgfb*^ret/ret^ tumors). **J** Quantification of CD206^+^ macrophages, cultured with pericyte- or endothelial cell-primed or mock medium. Either IL4, IFNγ, or no cytokine was added to the cultures. Analysis was performed 48 h post addition of primed media. Median with 95% CI is shown. *Pdgfb*^ret/+^ cell co-cultures are depicted in black (*n* = 3) and *Pdgfb*^ret/ret^ in red (*n* = 3). Two-sided *t*-test. Median is indicated (**B**, **C**, **E**, **G**, **J**). *\*p* < 0.05, *\*\*p* < 0.01, *\*\*\*p* < 0.001, ns not significant. Source data are provided as a Source data file.

clearly upregulated in MES II cells only, ISS appeared to be moderately up in both MES II and the PC cluster, the latter representing stromal cells with mesenchymal features. When we investigated the spatial context of the sub-signatures, we observed HSS to be mainly expressed around hypoxic zones, whereas ISS expression appeared mainly in the distant, tumor infiltration zone (Fig. 6C). RCTD estimated cell type proportions corroborated these findings, as we observed MES I and MES II cells to contribute equally in hypoxic zone capture spots. Conversely, we found a majority of the tumor invasive zone spots to feature a high proportion of MES II cells (Figs. 6D and S13A).

In summary, these results underline the unique character of MES II cells and imply that they are primed in hypoxic regions in contact with polarized macrophages, followed by the development of invasive features and permeation of the tumor periphery. Given the increased prevalence of MES II cells in *Pdgfb*^ret/ret^ tumors, we describe a mechanism triggered by the absence of pericytes, in which TME and glioma cells interlock, ultimately leading to a more effective infiltration of the surrounding brain parenchyma and aggravated tumor development.

### Pericyte depletion reinforces a highly invasive, *Met*-expressing subset of mesenchymal GBM cells

Having observed a higher abundance of polarized macrophages in *Pdgfb*^ret/ret^ gliomas and their location in mesenchymal tumor areas, we next aimed to investigate the functional dependency between tumor-supportive macrophages and invasive MES II cells in a pericyte-deprived microenvironment. Exploration of our scRNA-seq dataset with CellChat[26] revealed active HGF/MET signaling to occur exclusively between sending, *Hgf*-expressing macrophages and receiving, *Met*-expressing MES II cells in *Pdgfb*^ret/ret^, but not in *Pdgfb*^ret/+^ glioma tissue (Fig. 7A, B). *MET* is frequently over-expressed in human GBM due to amplification, mutation, and fusion events[78–81] and has been demonstrated to promote tumorigenesis through stimulation of glioma cell migration and invasion[82]. Indeed, when we determined the *Met* expression in our spatial transcriptomics dataset, we found *Met* to be expressed in the same area as the ISS sub-signature (Figs. 6C and 7C). Notably, we observed a very pronounced invasive pattern with a stRNA-seq sample that consisted of a smaller tumor core and a relatively large physiologic tissue area, including cortical zones: here, *Met* expression was almost completely absent in the *RCAS*^+^ tumor core zone, while it could be found in capture spots otherwise dominated by a non-tumorous expression profile (Fig. S14A, B).

To further validate the functional relationship between *Met*-expressing cells and macrophages, and to determine their occurrence in distinct glioma niches, we performed a combination of multiplex in situ hybridization and immunostaining on *Pdgfb*^ret/ret^ glioma sections, using probes against *Hgf*, *Aif1* (IBA-1), *Fosl1*, and *Met* as well as an antibody against PODXL. We frequently detected *Hgf*^+^/*Aif1*^+^ cells in the vicinity of *Fosl1* and *Met* expressing cells (Fig. 7D). Moreover, we also found cells that co-expressed *Fosl1* and *Met*, indicative of MES II cells (Figs. 7D and S14C). Notably, we observed an abundance of *Aif1*, *Met*, and *Fosl1* expressing cells not exclusively in hypoxic regions but also in the perivascular space of *Pdgfb*^ret/ret^ glioma sections, indicating a putative opportunistic occupation and exploitation of this glioma niche vacated by pericytes (Figs. 7D and S14C).

Due to the reported high tumorigenic and invasive potential of *Met*-expressing cells, and more permissive conditions of the TME upon pericyte depletion, we postulated that MET^+^ cells are among the main drivers of the observed aggravated tumor progression in *Pdgfb*^ret/ret^ mice. To investigate this hypothesis, we used flow cytometry to isolate one population of MET^+^, F4/80^−^ and one population of MET^−^, F4/80^−^ cells as a control from glioma tissue and expanded these cells in culture. Interestingly, MET^+^, but not MET^−^ cells proved to be difficult to expand ex vivo and exhibited a significantly lower sphere-induction potential when compared to control cells, implying the dependency of these cells on the glioma microenvironment for self-renewal (Fig. 7E). Indeed, the significant growth and survival advantage of MET^+^ cells in the presence of HGF illustrated their dependency on this pathway (Fig. 7F). In line with this proposition, and in keeping with their mesenchymal phenotype, upon intracranial transplantation, we found MET^+^ cells to generate highly aggressive, 100% penetrant gliomas, that developed significantly faster than transplanted MET^−^ cells (Fig. 7G), irrespective of injection into *Pdgfb*^ret/+^ or *Pdgfb*^ret/ret^ animals (Fig. S14D). When we compared tissue sections from MET^+^ and MET^−^ gliomas, we observed a strong expression of the glioma MES-like marker CD44[56], and the abundance of hypoxic areas in MET^+^ tumors, confirming the mesenchymal character of these gliomas (Fig. 7H).

Taken together, our studies suggest that pericytes attenuate alternative macrophage activation and interfere with their stimulation of an extreme mesenchymal malignant cell subpopulation that drives tumorigenesis and glioma cell invasion.

### HSS and ISS expression is associated with poor survival and spatially linked to glioma cell infiltration and the absence of pericytes

To find further support for a link between the absence of pericytes and a more aggressive GBM phenotype characterized by increased invasion into the surrounding brain parenchyma, we analyzed public patient cohorts with clinical follow-up and stRNA-seq data. First, we assessed the correlation between the MES II signatures derived from our scRNA-seq data with the outcome in the TCGA High Grade Glioma (HGG) cohort[83,84]. Indeed, patients with above-median expression of either MES II HSS, ISS, or the top 25 DEG of cluster MES II exhibited a significantly shorter progression-free interval (PFI), as well as overall

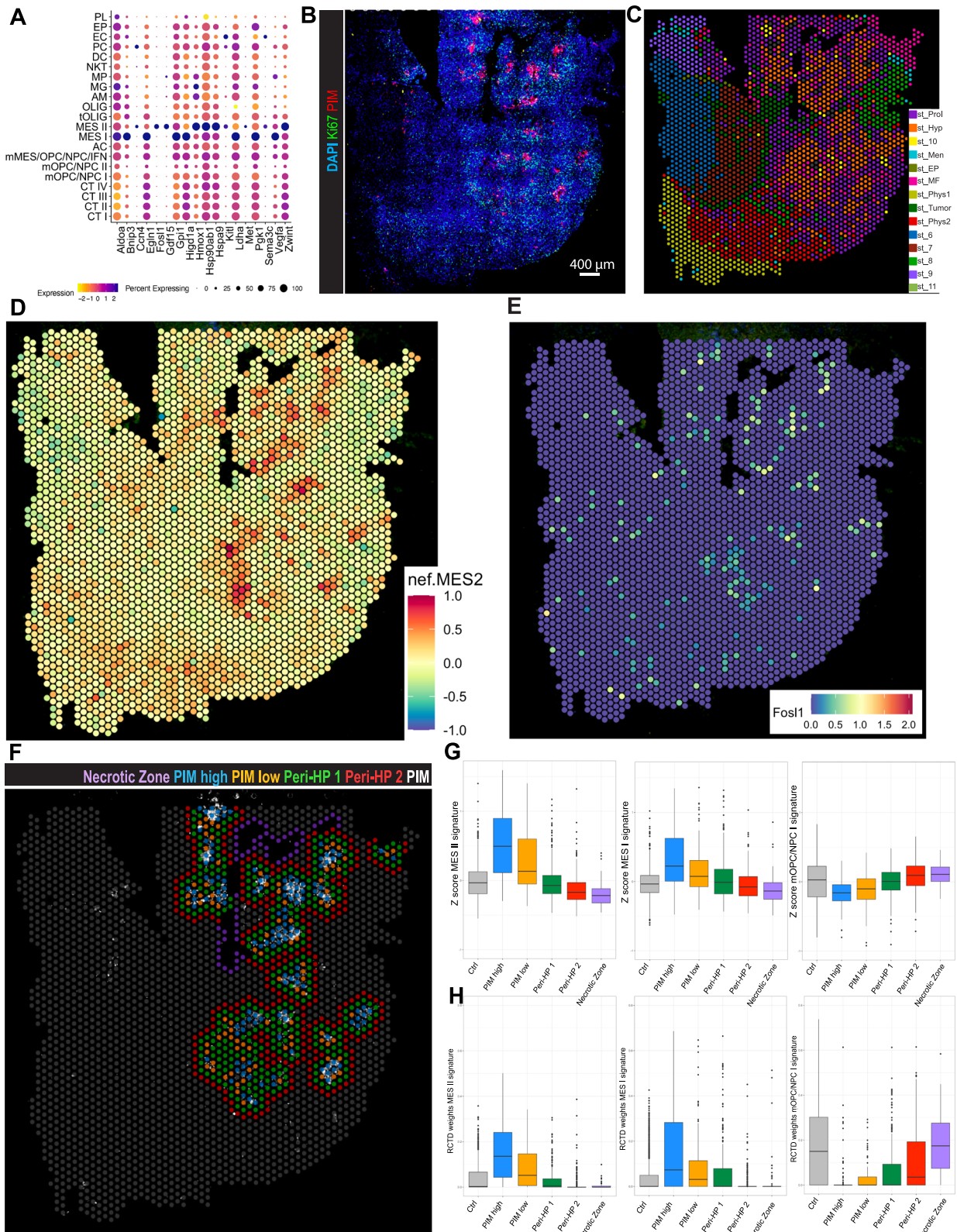

survival (OS) (Fig. S15A, B). Intriguingly, a core pericyte gene expression signature (*RGS5*, *KCNJ8*, *HIGD1B*, and *NDUFA4L2*)[25] was expressed at a lower level in recurrent samples from the cohort, compared to the matched primary tumors, in keeping with previous findings that recurrent GBMs exhibit a more aggressive growth pattern with a more pronounced MES signature[3,85] (Fig. 8A). Next, we aimed to determine the spatial localization of the MES II signature in human GBM samples.

To that end, we explored a GBM atlas of spatially resolved transcriptomics that were histologically classified applying the Ivy-GAP database classification system[86] (Fig. S15C). In accordance with observations from our HGG mouse model, a module score analysis revealed higher expression of the MES II/ HSS genes in the mesenchymal tumor zone, characterized by high MES2[44] signature expression ("vascular zone" in Ravi et al.) (Figs. 8B and S15C–E). Conversely, the

**Fig. 5 | Localization and functional characterization of mesenchymal cell subgroups with spatial transcriptomics analysis. A** Dot plot showing the scaled expression of selected MES I and MES II differentially expressed genes (combined *Pdgfb*<sup>ret/ret</sup> and *Pdgfb*<sup>ret/+</sup>) across all scRNA-seq clusters. **B** Representative image of a *Pdgfb*<sup>ret/ret</sup> glioma section (stRNA-seq sample #1159) mounted onto the capture area of a Visium Gene Expression Slide (VGES), and immunostained against PIM and Ki67. **C** Image of a *Pdgfb*<sup>ret/ret</sup> glioma section (#1159), mounted on a VGES capture area. 14 spatial clusters were annotated based on the integration of all 8 samples (four *Pdgfb*<sup>ret/ret</sup> and four *Pdgfb*<sup>ret/+</sup> glioma sections). **D** Image of a *Pdgfb*<sup>ret/ret</sup> glioma section (#1159) colored by the mean *Z*-score of the MES2 signature from Neftel et al.

**E** *Fosl1* expression in stRNA-seq sample #1159. **F** VGES capture area of stRNA-seq sample #1159, showing a manual annotation of a hypoxic gradient. Hypoxic zones are highlighted by PIM positivity ("PIM high" and "PIM low"), while peripheral zones are PIM negative ("Peri-HP1" and "Peri-HP2"). **G, H** Prevalence of MES II and mOPC/NPC I cells per spot in manually annotated hypoxic and peri-hypoxic zones defined in (**F**). All remaining capture spots served as a control zone. Box plots show the median (center line) of the average *Z*-score per spot (**G**) and RCTD proportions (**H**), the 25th and 75th percentiles (bounds of box), and whiskers extending to data points within 1.5 × the interquartile range; points outside this range are shown as outliers. Source data are provided as a Source data file.

MES II/ ISS genes appeared to be more highly expressed in zones that were histopathologically characterized as "infiltrative" and "white matter" (Figs. 8B and S15C, D, F). Finally, we used the core pericyte marker gene signature as a proxy for pericyte abundance, and found it to be upregulated in the "cellular" tumor zone (Fig. 8C). In contrast, lower pericyte marker gene expression was observed in the "vascular"/MES2 high tumor zone, and was virtually absent in the infiltrative tumor areas (Fig. 8C), indicating that the abundance of pericytes and HSS/ISS-expressing mesenchymal glioma cells are anti-correlated.

Collectively, we propose a model in which the relative absence of pericytes in recurrent human GBM instigates immune-suppressive macrophages to support a pattern of perivascular invasion of glioma cells with pronounced mesenchymal features, leading to reduced progression-free and OS.

## Discussion

The perivascular space has been elaborately characterized as a microenvironment that drives progression and invasion of human gliomas[5,87,88]. Here, we provide evidence that cells embedded within the vascular basement membrane – pericytes – mitigate the tumor-supportive nature of the PVN through crosstalk with endothelial cells and macrophages, primarily affecting the glioma molecular phenotype, and secondarily cell proliferation and migration, consequently decelerating tumor progression (Fig. 9).

Pericytes have been described to exert pro-tumorigenic functions by contributing to the establishment of immune-tolerance, acting on the ECM composition, and driving epithelial-to-mesenchymal (EMT) transition processes[17,85,89,90]. Here, we unexpectedly identified pericytes as the most active signaling cell type in relative terms in both human and murine gliomas. We applied a profoundly characterized mouse model that features the most extensively pericyte-deficient brain microvasculature compatible with adult life observed so far, to study the consequences of pericyte deprivation on glioma development[14,22,91]. Pericyte depletion led to an aggravated course of tumor development, suggesting a more complex role of pericytes in glioma than described in the literature hitherto. Importantly, the phenotype was observed in two distinct tumor models, including one that is independent of potentially confounding oncogenic PDGF-BB signaling. The notion of a previously underestimated complexity in the PVN of gliomas is corroborated by the recent identification of different types of perivascular fibroblasts that might occupy the PVN in addition to, or instead of, pericytes[21,92,93], and can have been missed by previous studies. Although the different subtypes of mesenchymal perivascular cells still lack adequate characterization based on conclusively ascertained marker panels, it is conceivable that fibroblasts and pericytes might affect glioma cells and their microenvironments in different ways.

Pathophysiological alterations of the brain vasculature, such as blood-brain barrier dysfunction and neuroinflammation associated with increased immune cell infiltration in the brain, have been observed in neurodegenerative diseases[94,95], SARS-CoV-19 infection[96,97], and demonstrated to be plausibly related to a loss of pericytes in various mouse models featuring a diminished pericyte:endothelial cell ratio[14,98]. Furthermore, the generation of subcutaneous tumors in *Pdgfb*<sup>ret/ret</sup> mice led to a more defective tumor vasculature with a hypoxic microenvironment and increased trafficking of MDSCs to the tumor site[51]. In this study, we did not detect exacerbated hypoxia in pericyte-deprived tumors, but identified altered endothelial cell signaling in the context of an aggravated vascular dysmorphogenesis as a potential driver of glioma progression. One aspect of how the pericyte-poor glioma vasculature fuels tumor growth is the increased number of vessel co-opting tumor cells infiltrating the brain parenchyma. This invasion mechanism, reminiscent of OPC-endothelial migration[99], has been linked to OLIG2⁺/*Wnt7b*⁺ tumor cells[50], which is in line with our results indicating OLIG2-positive, OPC-like glioma cells as predominant tumor cells in our model system, and the observed increases in WNT signaling of this cell compartment in *Pdgfb*<sup>ret/ret</sup> tumors (Fig. S8C). Notably, the infiltrative, vessel-associated glioma cells that we observed could potentially acquire mesenchymal features: a recent study identified a subset of highly vessel co-opting glioma cells, residing midway between the OPC/NPC-like and the mesenchymal cell state on the transcriptomic axis, and linked the induction of this invasive state to angiocrine factors from brain blood vessels[100]. Moreover, our findings are also relevant in light of recently published data demonstrating an increased glioma cell invasion in recurring IDH-wild-type gliomas, known to be pericyte-deprived, that was linked to an altered tumor microenvironment[4].

In accordance with Hong et al.[51], we also measured a higher influx of MDSCs in pericyte-deprived gliomas, which we found paired with an increased attraction of macrophages that had developed differential polarization phenotypes and could be stratified into subgroups. Spatially, our data showed different preferences of glioma-associated macrophages for the perivascular (*Mrc1*/*Cd206*⁺ macrophages) and hypoxic (*Msr1*/*Cd204*⁺ macrophages) niche based on marker gene expression profiles, implying differential activation of tumor-supportive gene programs in the tumor microenvironment; a notion also supported by previous studies[4,24,101].

The abundance of alternatively activated macrophages has not only been shown to be associated with the MES subtype, but in fact to induce it[56] in a manner critical for gliomagenesis and therapy resistance[24,58,102]. Here, by integrating scRNA-seq, stRNA-seq, and multiplex imaging data, we identify hypoxic, macrophage-dense tumor zones as the predominant niche for mesenchymal glioma cells. We find that whereas most MES tumor cells appear bound to the hypoxic niche, a specific group of extreme MES cells express invasion-related genes, potentially empowering them to leave the hypoxic niche, migrate towards the tumor periphery, and infiltrate the physiologic brain parenchyma. Indeed, in our stRNA-seq data, we observed an increased expression of invasion-related genes at the tumor leading edge. This observation of a high degree of glioma cell infiltration in non-malignant brain parenchyma, emanating from a driving hypoxic/mesenchymal core zone, and advancing through adjacent tiers of proliferating and OPC/NPC/AC-like tumor cells, confirms a recent report on the spatial organization of GBM[103].

Furthermore, we show that MET signaling, a key mesenchymal and cell motility pathway, is activated exclusively in extreme mesenchymal tumor cells of pericyte-poor tumors, and fueled by

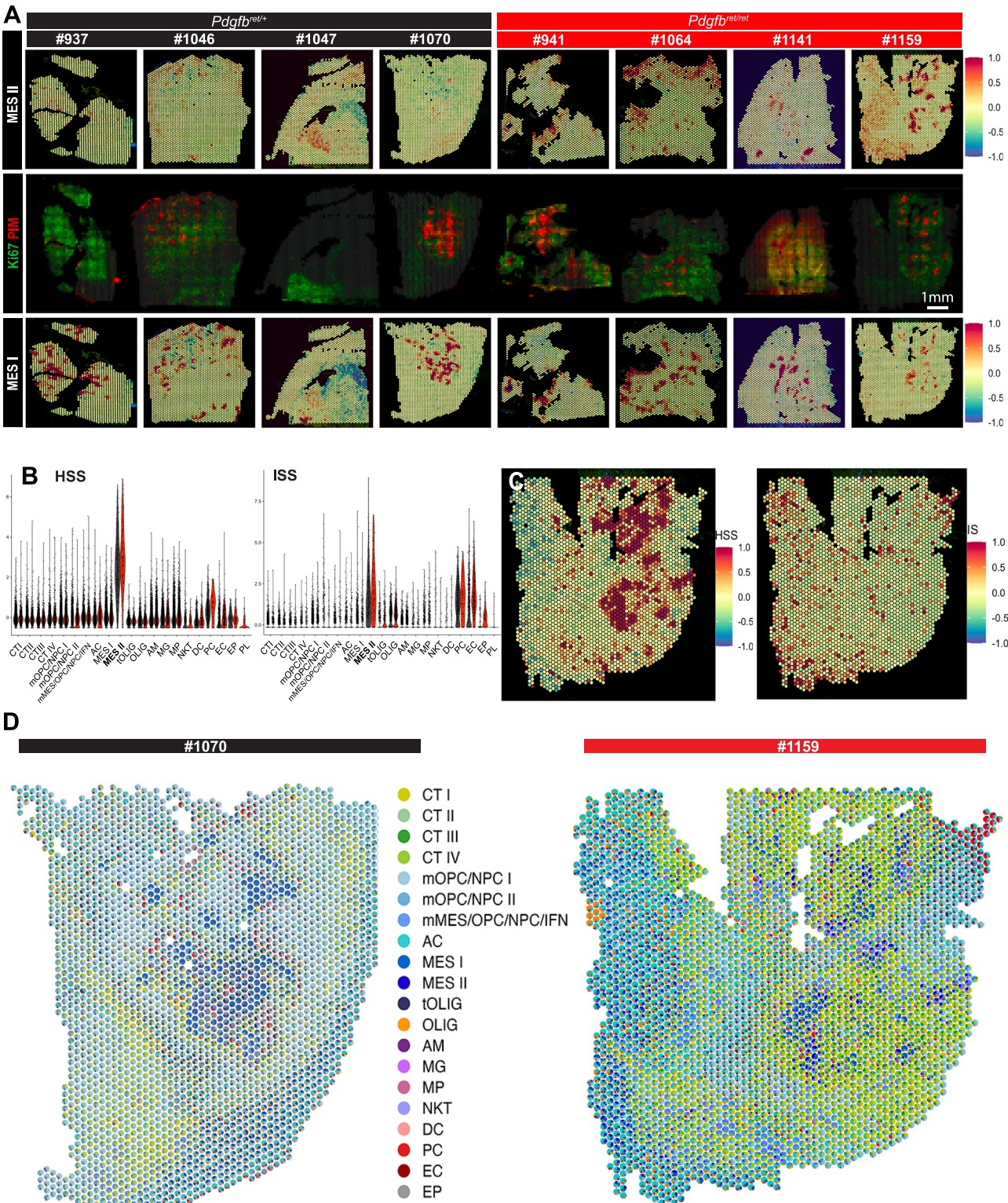

**Fig. 6 | Mesenchymal tumor cells, differing in distinct gene expression programs, prevail in spatially separated glioma niches. A** VGES capture area images of 4 *Pdgfb*ret/+ (black label) and 4 *Pdgfb*ret/ret (red label) glioma sections (upper and lower panels), colored by the mean *Z*-scores of the scRNA-seq clusters MES I and II. The middle panels show the corresponding immunostainings of PIM and Ki67 for each section. **B** Violin plots showing the signature score of HSS (hypoxia and stemness) and ISS (invasiveness) across all scRNA-seq clusters for *Pdgfb*ret/ret and *Pdgfb*ret/+ tumor-derived cells. *Pdgfb*ret/+ is depicted in black and *Pdgfb*ret/ret in red. **C** Image of a *Pdgfb*ret/ret glioma section (#1159) colored by the signature score of HSS (hypoxia and stemness) and ISS (invasiveness). **D** Spatial scatterpie chart of glioma sections #1070 (*Pdgfb*ret/+) and #1159 (*Pdgfb*ret/ret), showing the RCTD-inferred proportions of different cell types per spot. Source data are provided as a Source data file.

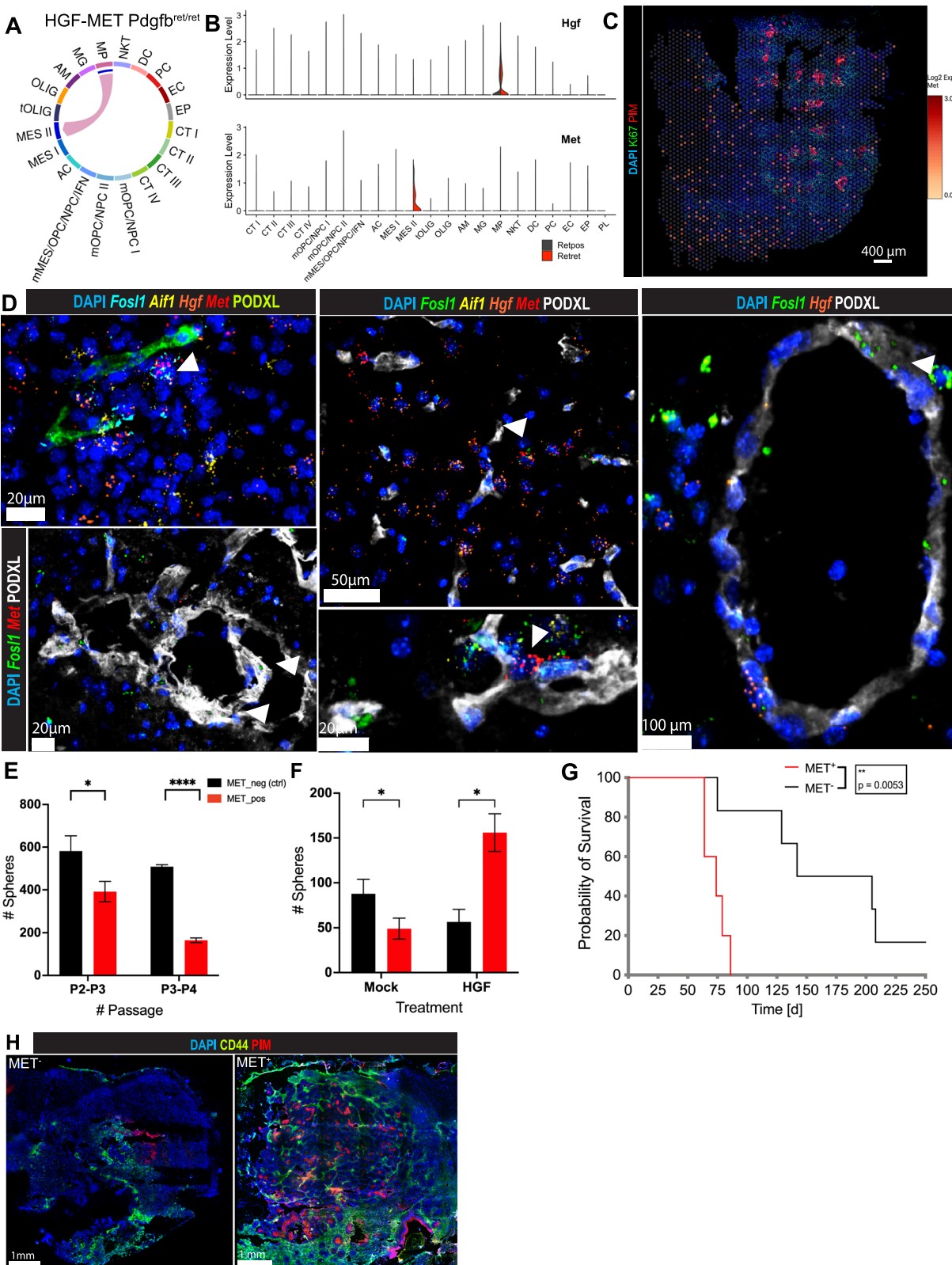

glioma-associated macrophages that express the ligand HGF, marking a shift in the generally hardwired OPC-like, PDGFB-driven RCAS/tv-a murine glioma model[36] toward the mesenchymal subtype upon pericyte deprivation. Interestingly, MET activation has already been described for a pericyte-depleted mouse model of breast cancer, and poor pericyte coverage in combination with high *MET* expression has been identified as a predictor of poor outcome in patients with

invasive breast cancer[104]. However, we do not share the observation of significantly increased hypoxia upon pericyte depletion but surmise instead a remodeled microenvironment as the underlying cause of MET activation. Moreover, AP-1 transcription factor and major regulator of mesenchymal features, FOSL1, a marker of the extreme mesenchymal cells in our dataset, and macrophage-secreted HGF, could both potentially induce *Met/MET* expression[59,104–106].

**Fig. 7 | Mesenchymal tumor cells interacting with macrophages drive an invasive glioma phenotype. A** Chord diagram of the HGF-MET signaling pathway, active exclusively in *Pdgfb^ret/ret* tumors, based on the CellChat analysis. The plot shows the *Pdgfb^ret/ret* tumor-specific interaction of macrophages and MES II cells. **B** Violin plot shows the logNormalized expression of *Hgf* and *Met* in *Pdgfb^ret/ret* and *Pdgfb^ret/+* cells across all scRNA-seq identified clusters. **C** Spatial distribution of *Met* expression for a representative *Pdgfb^ret/ret* stRNA-seq sample (#1159). **D** Combined multiplexed immunostaining-in situ hybridization (ISH) analysis for the detection of *Hgf*, *Aif1*, *Fosl1*, *Met*, and PODXL in PDGFB-induced gliomas, derived from *Pdgfb^ret/ret* mice. Arrowheads indicate *Fosl1⁺* and *Fosl1⁺/Met⁺* cells. **E, F** In vitro sphere-induction analysis of MET⁺, F4/80⁻, and MET⁻, F4/80⁻ cells, isolated from PDGFB-induced glioma tissue. Cells were seeded at passage (P) 2. Total sphere number per culture was determined at passages P3 and P4 (**E**). Cells seeded at P6 were treated with HGF or mock (**F**). MET⁻ cells are depicted in black and MET⁺ cells in red. Two-sided *t*-test. Mean (bars) with SD is shown. Three biological replicates per treatment were analyzed. **G** Kaplan–Meier curves showing symptom-free survival of mice transplanted with MET⁺ (5) and MET⁻ (6) glioma cells. Log-ranked test. **H** Representative immunostainings of CD44 and PIM on glioma sections derived from intracranial engraftment of MET⁺ and MET⁻ glioma cells. Source data are provided as a Source data file.

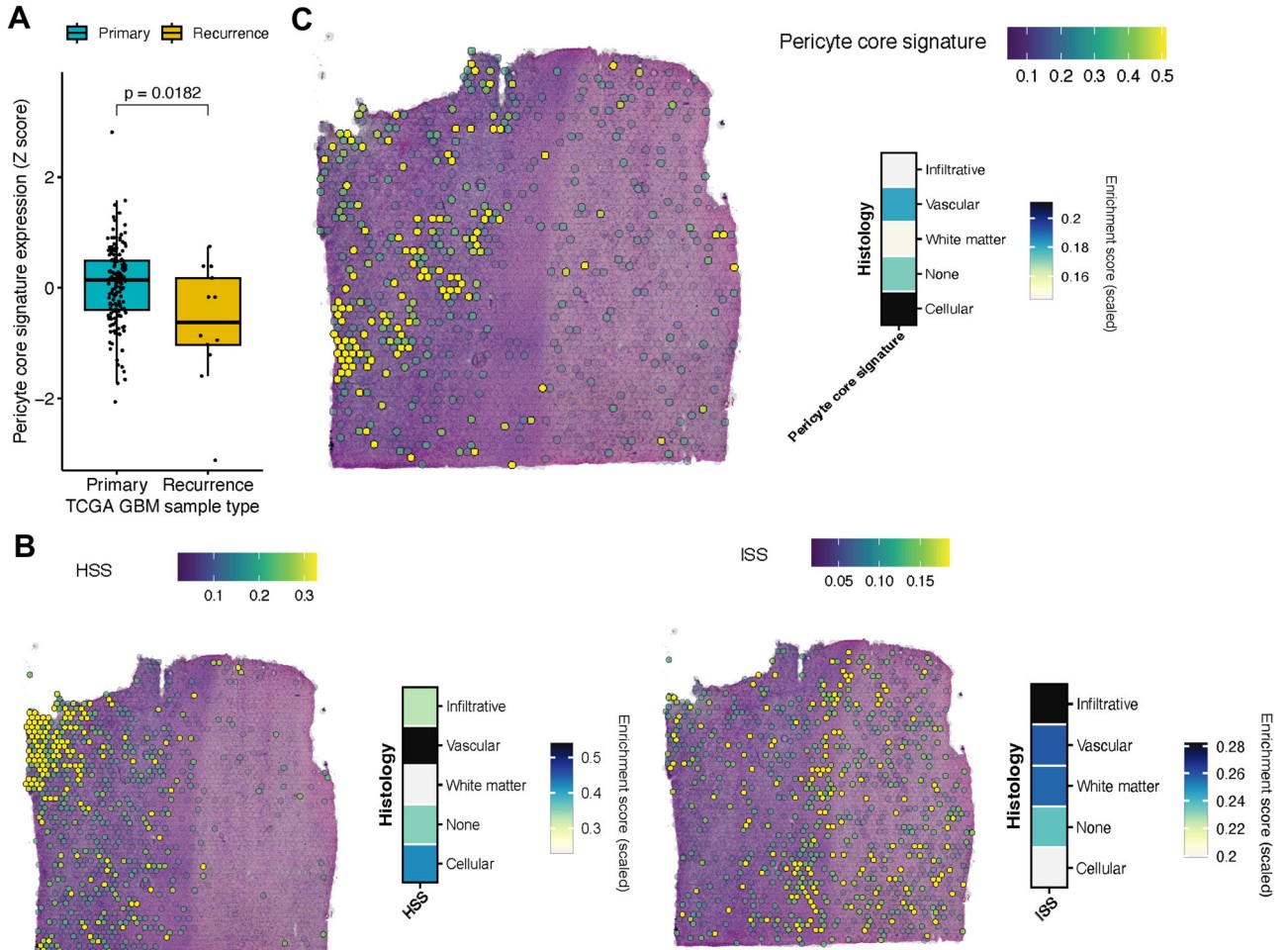

**Fig. 8 | Spatial localization analysis of GBM patient datasets. A** Box plot showing the expression of the pericyte core signature in primary (*n* = 153) and recurrent (*n* = 13) TCGA GBM samples. Two-sided Wilcoxon test. Spatial plots and heatmaps showing HSS (**B**, left) and ISS (**B**, right), and pericyte core gene module scoring (**C**) in a spatially resolved transcriptomics Human GBM specimen[86].

The enhancement of mesenchymal features of a subset of GBM cells triggered by alterations of the TME was the most striking alteration upon pericyte depletion in the glioma mouse model that we analyzed. In addition, we also observed subtle changes in tumor composition, such as an expansion of non-tumor oligodendrocyte-progenitor cells, a more pronounced NPC-like cell compartment, and an increase of vessel co-opting OLIG2⁺ glioma cells at the tumor leading edge, as well as more dramatic alterations, like the higher abundance of polarized myeloid cells. In this regard, it is interesting to note that longitudinal shifts towards a higher non-tumoral OPC and NPC content and a stronger expression of NPC-like signature genes towards the leading-edge, increased tumor cell infiltration and upregulation of immune-suppressive, myeloid-specific expression profiles

have been reported for recurring IDH-wild-type gliomas, were found independent of general subtype switches, and could be related to the underlying physical structure and microenvironment of the tumor[4,85]. Taking also into account that the MES-subtype appears to dominate in recurring gliomas[3,85], our data suggest certain parallels regarding the mode of tumor progression between recurrent and pericyte-deprived high-grade gliomas.

In this study, we observed a faster and more aggressive tumor progression in glioma models upon pericyte depletion and subsequent TME remodeling. Our results stress the significance of the TME for the course of glioma progression and its therapeutic potential despite numerous challenges for treatment design due to its complexity. However, they also imply the induction of significant changes in

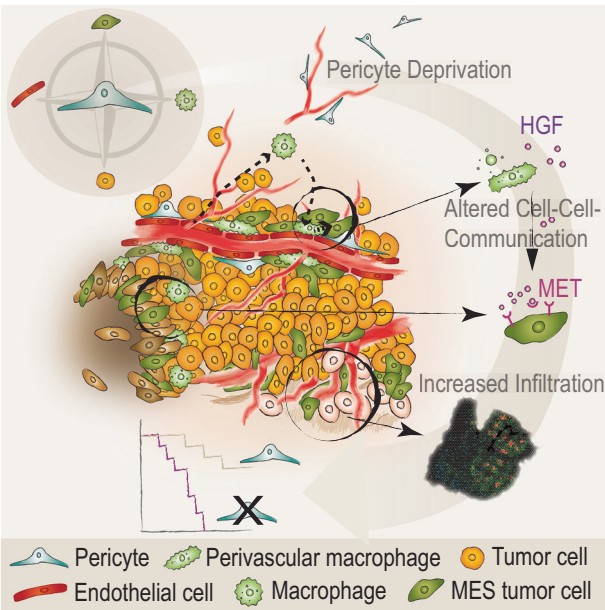

**Fig. 9 | Graphical abstract.** Pericyte deficiency remodels the glioma TME, aggravating tumor development through reinforced tumor cell invasion and immune suppression.

glioma dynamics upon unspecific targeting of pericytes, and consequently demand caution when considering an overall therapeutic ablation of perivascular cells, a concept which is being debated[20,94,107]. It has been shown that glioma stem-like cells bear the capability to generate perivascular stromal cells that could potentially be used as drug targets[19,20]. However, it is notoriously difficult to determine and validate the identity of pericytes, due to their heterogeneous origin, marker expression, and their entanglement with adjacent cells, making it conceivable that other vessel-associated TME cells than pericytes are glioma cell-derived and exert tumor-supportive functions[13,16,108].

It is noteworthy that the measured phenotypic shifts towards increased glioma cell invasion and the adoption of the MES cell state are reminiscent of the infiltrative glioma growth pattern that developed upon anti-angiogenic treatment in several studies[109–111]. To avoid such phenotypic drift, a powerful future integrative treatment approach, including the glioma vasculature, requires profound knowledge of the diverse subtypes and roles of perivascular stromal cells as a prerequisite to effectively target putative glioma-supporting perivascular cells. At the same time, our results stress the therapeutic importance of preserving pericytes that are contributing to vessel maintenance processes, a notion that is also corroborated by previous findings characterizing pericytes as vulnerable to radiation treatment and linking their ablation to radiation-induced necrosis[112], an unfavorable condition that typically induces a mesenchymal switch in glioma cells, making them fit to escape and invade[113]. Finally, it is noteworthy that pericytes are involved in vessel fortification, a beneficial process superior to mere vessel depletion during therapeutic normalization of the glioma vasculature[114]: the anti-angiogenic drug bevacizumab was shown to increase vessel pericyte coverage and chemotherapy response in breast cancer[115] and to reduce necrosis in a combination treatment with lomustine[114].

Overall, our study reveals the multifaceted nature of pericytes and their role in mitigating high-grade glioma development by orchestrating a tumor-suppressive microenvironment. To achieve improvements in glioma therapy, future studies need to elucidate how specific subgroups of perivascular stromal cells influence the glioma microenvironment and aim to therapeutically target tumor-supportive

perivascular cells, while actively shielding pericytes that counteract glioma.

## Methods
### Generation of murine brain tumors
All animal experiments were performed according to institutional guidelines and approved by the local ethics committee in Lund (permit numbers M167/15, 14122/2020), and all experiments were conducted using a balanced number of male and female subjects bred on a mixed background including FVB and C56BL/6. Transplantations of 100k Gl261, 50k $TpS3^{-/-}$, $H\text{-}Ras$ over-expressing murine glioma cells or 50k MET+ or MET− glioma cells into the right frontal brain lobe of 2–6 months old mice were carried out. For IVIS experiments, 50k $TpS3^{-/-}$, $H\text{-}Ras$ cells expressing a bicistronic GFP-luciferase sequence were transplanted. The RCAS/tv-a system used in this study has been described previously[116–119]. *Nestin* Ntv-a, $Pdgfb^{ret/+}$, and *Nestin* Ntv-a, $Pdgfb^{ret/ret}$ mice were used for the RCAS-mediated gliomagenesis in this study. After stereotactical injection of 100k DF-1 virus-producing cells (1:1 mix of RCAS-shp53- and RCAS-PDGFB-transduced cells), mice were observed until they developed neurological and/or physical symptoms related to tumor development, and euthanized. The Mantel-Cox test was applied to test for statistical significance in survival between the different cohorts.

### Immunostaining
For immunostaining, tissues were fresh frozen in OCT Cryomount (Histolab). 10 µm tissue sections were dried at room temperature and fixed in ice-cold acetone for 10 min. All of the following steps were performed in a humidified chamber. After washing in PBS, sections were blocked with serum-free blocking reagent (DAKO, X090930-2) for 90 min at room temperature. Primary antibodies were diluted 1:100 in PBS + 1% BSA, and sections were incubated overnight at 4 °C. After washing with PBS, secondary antibodies (1:1000 in PBS + 1% BSA) were applied for 1 h at room temperature. Sections were washed and mounted in DAPI-free or DAPI-containing mounting medium. If not stated otherwise, microscopic pictures were taken with an Olympus BX63 microscope using the cellSens software package and a 20× objective. Image postprocessing and analysis were done with Qupath (v0.4.3)[120]. Minimal gamma corrections were used to enhance the visibility of features and improve the print quality.

ISH/IF with fresh frozen tissue sections was performed with an RNAscope Multiplex Fluorescent v2 Assay combined with Immunofluorescence - Integrated Co-Detection Workflow according to protocol #MK 51-150/Rev B/Date 02112021. OPAL dyes were applied for fluorescence labeling. The sections were imaged at 40x with an Akoya PhenoImager using PhenoChart (v1.0.12).

To determine the pericyte vessel coverage rate, murine glioma sections were immunostained for ACTA2 or PDGFRβ, respectively, and PODXL. 22 fields of view (FOV) of three $Pdgfb^{ret/ret}$ and four $Pdgfb^{ret/+}$ animals were analyzed for Gl261-derived gliomas, and 200 FOV of three $Pdgfb^{ret/ret}$ and five $Pdgfb^{ret/+}$ animals were analyzed for PDGFB-induced gliomas. Pericyte coverage was calculated by dividing the total ACTA2 or PDGFRβ positive area by the total PODXL positive area, applying Angiotool (version 0.5)[121] and Fiji (v2.9.0)[122]. An unpaired t-test was used to test for statistical significance between the different cohorts.

### Characterization of the murine glioma tissue
To assess vessel parameters, total area, length, and junctions, glioma sections of seven $Pdgfb^{ret/ret}$ and eight $Pdgfb^{ret/+}$ mice were immunostained for PODXL and PECAM1, and 9-10 FOV per mouse were analyzed using Angiotool (version 0.5)[121]. The following Angiotool parameters were applied: vessel diameter- 7; threshold- 20-255, small particle filter- 350. Junctions are defined as points in the generated image where two or more skeleton segments meet.

For vessel perfusion analysis, mice were briefly sedated with isoflurane and injected with 40 μl Oregon-Green Dextran (70 kDa, 25 mg/ml, ThermoFisher Scientific) in a retro-orbital fashion. After a 15 min incubation, animals were subjected to cardiac perfusion with 10 ml PBS and 10 ml 4% pFA, and brains were collected subsequently. For quantification, glioma cryosections from various tumor sites of two $Pdgfb^{ret/ret}$ and two $Pdgfb^{ret/+}$ mice were stained for PECAM1, and PECAM1- as well as Oregon-Green- positive areas were quantified, and ratios determined.

For hypoxia and proliferation analysis, glioma sections were immunostained against HIF2α and Ki67, counterstained with DAPI, and the HIF2α/DAPI and Ki67/DAPI total area ratios were determined. In the case of the HIF2α quantification, OLIG2 immunostaining served as a proxy to determine the tumor boundaries and infiltrative zones.

For hypoxia analysis with the Hypoxyprobe system (hpi), animals were injected in a retro-orbital fashion with 50 μl pimonidazole (60 mg/kg body weight), euthanized after 30 min of incubation, and brains collected.

To analyze the invasive rim (IR) zone, glioma cryosections of three $Pdgfb^{ret/ret}$ and five $Pdgfb^{ret/+}$ mice were immunostained for OLIG2 and PECAM1, and evaluated subsequently. Three to seven FOV/section were analyzed by assessing the border of the tumor bulk and counting of OLIG2$^+$ cells, PECAM1$^+$ cells, OLIG2$^+$ cells in direct contact with PECAM1$^+$ cells, as well as determining the total PECAM1$^+$ area in the adjacent IR zone with Fiji[122].

## In vivo bioluminescence imaging of glioma development in mice

Murine $p53^{-/-}$ and $H$-$Ras$ over-expressing glioma cells were transduced with a CMMP vector (provided by Laurent Roybon, Lund University, Lund, Sweden) containing the luciferase cDNA sequence. Three- to six-month-old $Pdgfb^{ret/ret}$ and $Pdgfb^{ret/+}$ mice were subjected to stereotactical engraftment of 100k luciferase over-expressing glioma cells into the right frontal brain lobe. From 5 days after engraftment, the tumor growth was monitored by non-invasive 2D bioluminescence (BLI) imaging, using IVIS-CT spectrum (PerkinElmer) and Living Image Analysis Software (v4.7.3, PerkinElmer). Briefly, mice were anesthetized with 3% isoflurane gas and injected intraperitoneally with 150 mg D-Luciferin/kg of body weight (Revvity) in PBS prior to imaging. Acquisition of 2D images was performed sequentially with a 5-min interval between different segments of exposure (emission: open filter, f/stop: 1, binning: 8). BLI signal intensity was quantified in total flux (photons/s) after deducting the average background signal (Bkg) from measurement regions of interest (ROI), using the Living image analysis software (v4.7.3, PerkinElmer). To estimate tumor growth kinetics, the data were fitted to a non-linear model (least squares fit).

## Cell culture

All cells were cultured at 37 °C and 5% CO$_2$, and were frequently checked for mycoplasma infections using the MycoAlert™ Mycoplasma Detection Kit (Lonza). Unless stated otherwise, all media were supplemented with glutamine (2 mM, Corning), Penicillin/Streptomycin (50 μg/ml, Corning), HEPES (25 mM, Corning), and 10% FBS (Corning). DF-1 cells were cultured in DMEM high glucose (Corning). Macrophages were cultured in RPMI 1640 glucose (Corning) supplemented with 50 ng/ml M-CSF (Miltenyi). Glioma cells were cultured under serum-free conditions as spheres, in medium supplemented with B27 (ThermoFisher Scientific, 17504044), and rhFGF (10 ng/μl, R&D, 233-FB-025), and hEGF (10 ng/μl, R&D, 236-EG-200) in DMEM/F12 (Gibco). Pericytes were cultured in pericyte medium containing relevant growth factors (3H Biomedical, 1201). Endothelial cells were cultured in endothelial cell medium (PromoCell, C-22020).

## TSP assay

2000 cells per 2 ml medium were seeded in 6 wells in triplicates. One week after incubation, spheres were counted with a Zeiss Axio Vert.A1 system. Subsequently, spheres were digested with Accumax (Sigma-Aldrich), triturated, and cells were counted with a Buerker chamber. 2000 cells per triplicate were seeded again for continuous analysis.

For the HGF-enriched TSP culture, glioma cells were incubated with recombinant murine HGF (50 ng/ml, R&D, 2207-HG-025) for 1 week.

## Macrophage assays with pericyte-conditioned medium and GAS6

For macrophage culture, bone marrow cells were extracted by crushing femora, tibiae, and ilia of female, 4–12 weeks old FVB/n mice in ice-cold PBS containing 10% FBS, and subsequently passing the suspension through a 70 μm filter. After centrifugation for 5 min at 350 g, the cell pellet was resuspended in NH$_4$Cl buffer for 5 min at RT, and then the cells were centrifuged again, followed by seeding of approximately $3 \times 10^5$ bone marrow cells per well (6-well plates) in macrophage medium. Bone marrow cells were differentiated into macrophages for 7–9 days.

For the conditioned medium assay, pericytes and endothelial cells at 80% confluency were cultured in macrophage medium w/o growth factors for 48 h. Subsequently, the medium was collected, centrifuged, filtered, and applied to macrophages at 50–80% confluency in 6-well plates. M-CSF (Miltenyi) was added to all wells, and IFNγ (14 ng/ml, PeproTech) or IL4 (14 ng/ml, PeproTech) was added to the respective sample wells. After medium exchange, the macrophages were cultured for 48 h, scraped off, and frozen at −150 °C in cryopreservation medium (70% medium, 20% FBS, 10% DMSO). For the GAS6 assay, GAS6 (200 ng/ml, R&D) was added to the respective wells, and the macrophages were cultured for 72 h. Subsequently, the cells were fixed in a 1:2 mix of ice-cold 4% pFA and FACS buffer for 3 h.

## Glioma sample collection

All animals were monitored daily for neurological and physical symptoms, and sick animals reaching the defined breaking point were euthanized in accordance with the ethical permit. Terminally ill animals were sedated with isoflurane, decapitated, and the brains removed. Brains were immediately brought on ice, and a coronal cut through the tumor lesions was performed. Tumor material was then isolated and washed in ice-cold DPBS. Subsequently, the tumor tissue was dissociated using the Adult Brain Dissociation Kit (Miltenyi), resuspended in cryopreservation medium, and stored at −150 °C.

## FACS analysis

Cells were thawed quickly in a 37 °C warm water bath, resuspended in medium, and centrifuged at 220 g. Subsequently, cells were washed twice with 10 ml FACS buffer (DPBS with 5% FBS or BSA), centrifuged at $220 \times g$, and resuspended in 50–100 μl FACS buffer at a concentration of 1:100 per respective antibody. After 30 min incubation on ice in the dark, the cells were washed again in FACS buffer, passed through a 40 μm cell strainer (VWR), spun at $220 \times g$, and resuspended in FACS buffer at a density of 1–5 × 10$^6$ per 0.5 ml. The cells were then placed on ice and kept in the dark until analysis with a BD FACS Melody system (BD Biosciences) and an Aurora CS (Cytek), respectively. Post-FACS analysis of the recorded data was performed with FLOWJO (version 10.9.0, BD Biosciences). Sorted cells were centrifuged at 300 g, resuspended in 500 μl growth medium, and initially cultured in 24 wells. Gating strategies are shown in Fig. S16.

## Single-cell RNA sample preparation and sequencing

High-grade gliomas were initiated by stereotactically delivering the glioma oncogenic drivers PDGFB and $shp53$ into the SVZ of transgenic adult $Ntv$-$a$ mice that were either pericyte-poor ($Pdgfb^{ret/ret}$) or control ($Pdgfb^{ret/+}$). Seven $Pdgfb^{ret/+}$ and four $Pdgfb^{ret/ret}$ mice were sacrificed when displaying severe symptoms. Subsequently, whole tumors were

extracted and processed into single cell suspensions according to the protocol of the Adult Brain Dissociation Kit, mouse and rat (Miltenyi). In short, the whole tumor was minced into smaller slices with a scalpel and incubated for 30 min with the supplier's enzyme mixes, while being heated at 37 °C and mechanically grinded in the gentle MACS Octo Dissociater with Heaters (Miltenyi). Subsequently, the sample was resuspended and applied to a 70 μm MACS SmartStrainer (Miltenyi), after which the single cell suspension underwent subsequent steps for debris removal and lysis of erythrocytes. For library preparation, 18,000 cells/45 μl in 0.04% BSA in PBS were delivered to the Center of Translational Genomics at Lund University (CTG). Single-cell 3′ RNA-seq libraries were prepared using the 10x Chromium system (v3.1) by CTG. Samples were run using Read 1 28 cycles, i7 index 5 cycles, i5 index 0 cycles, Read 2 91 cycles on the Illumina NovaSeq 6000 to a depth of 50,000 reads/cell. This was performed in two different batches (time points). The first batch included two $Pdgfb^{ret/+}$ and one $Pdgfb^{ret/ret}$ mice, while the second batch included five $Pdgfb^{ret/+}$ and three $Pdgfb^{ret/ret}$ mice.

### Single-cell RNA data analyses

The raw reads were processed, mapped, and quantified using Cell Ranger count (v6.1.2)[123] with default settings, with an initial expected cell count of 10,000. We mapped the reads to a modified mouse reference genome (mm10) that contained the RCAS-PDGFB longest possible sequence following Cell Ranger instructions. Cell Ranger's filtered files were further filtered to keep cells with a number of transcripts lower than 50,000, a number of features larger than 250 and lower than 8000, and a percentage of mitochondrial and ribosomal genes lower than 10% and 25%, respectively. Finally, we filtered out mitochondrial and hemoglobin genes, as well as features with less than 20 counts across cells. We used DoubletFinder (v2.0.3) to filter out doublets, assuming a doublet formation rate of 7.5%. pN = 0.25 and pK were adjusted for each sample using pN–pK parameter sweeps and mean-variance-normalized bimodality coefficient (BCmvn) using paramSweep_v3[124]. We used DecontX (celda, v1.6.1) to filter out ambience DNA contamination[125]. We removed barcodes that had more than 25% of reads identified as ambient RNA contamination. QC and filtering steps were performed in every sample separately. After quality control, 16,086 cells were retained for the $Pdgfb^{ret/ret}$ batch (4 mice) and 42,838 cells were retained for the $Pdgfb^{ret/+}$ batch (7 mice). Data from both $Pdgfb^{ret/ret}$ and $Pdgfb^{ret/+}$ mice were integrated together to remove batch effects. Data analyses, including filtering, lognormalization, feature selection, scaling, dimensionality reduction (PCA and UMAP), and clustering were performed using Seurat (v4.0.3)[126–129]. Batch integration, specifying batch and sample[130], was performed using the Harmony algorithm v1.0. The Seurat function FindConservedMarkers was used for finding DEG between clusters. The top 50 DEG for every cluster were defined as the cluster/cell type signature. FindMarkers was used for testing for differential expression between conditions for every cluster separately. Cluster PL was excluded from DEG analyses as it was $Pdgfb^{ret/ret}$ specific. We used scCustomize (v2.1.2)[131] for visualization of the results, including UMAP, feature plots, and clustered dot plots of selected top DEG. scCustomize uses the k-means method for clustering dot plots. For the butterfly plot visualization, the AddModuleScore function in Seurat was used to calculate the Neftel et al. gene set enrichment/module scores, and the do_CellularStatesPlot function in SCpubr (v2.0.2)[132] was used to construct the scatter plots. Transcription factor activity analysis was performed on a downsampled (n = 20,000 cells) version of the dataset using DoRothEA[64–66]. Transcription factor regulons with the three highest confidence levels ("A", "B", and "C") were used, and the averaged activity scores per cluster were plotted using the do_TFActivityPlot function in the SCpubr package. We used default parameters for these tools unless otherwise specified.

### Z-score estimation and correlation matrix

To further explore gene expression patterns in our dataset and compare our defined gene signatures to human-defined counterparts, we used published available gene signatures from Neftel et al. and Couturier et al. Briefly, we used Neftel et al. gene signatures as reported by the authors: AC, OPC, MES1, MES2, NPC1, NPC2, G1S, G2M. The gene lists reported by Couturier et al. are from a continuous scale. Therefore, we selected the top 20 and 20 bottom genes of their eigenvectors DC1, DC2, and DC3, and defined 6 Couturier et al. signatures as DC1 top non-neuronal-like (DC1t/non-neuronal), DC2 top oligodendrocyte (DC2t/OLC), DC3 top progenitor (DC3t/progenitor), DC1 bottom neuronal-like (DC1b/neuronal), DC2 bottom mesenchymal-astrocyte-like (DC2b/Mes-Astro), and DC3 bottom non-progenitor (DC3b/non-progenitor). From these genes, we identified 1:1 orthologs between mouse and human using an orthology table downloaded from Ensemble (release 111) using the Biomart web-based tool. The exact genes included in the signatures are reported in the source data file. For every gene signature, the calculation of Z-scores was performed for each gene belonging to the signature. Z-scores enabled us to estimate how many standard deviations from the mean a value is. Considering that $x_{ij}$ is the expected value of gene $i$ in sample $j$, $\mu_i$ is the mean of expected values for gene $i$ across all $j$ samples, and $\sigma_i$ is the standard deviation of expected values for gene $i$ in all $j$ samples, the formula of the Z-score is:

$$Z_j = \left(x_{ij} - \mu_i\right)/\sigma_i$$

The signature score for every cell/spot is then calculated as the arithmetic mean of the Z-scores of all the genes included in the signature. We estimated pairwise Pearson's correlation coefficients for every pair of signature scores across all cells/spots. A correlation matrix was calculated and plotted using the corrplot R package (v0.92)[133]. The correlation matrix plot was ordered using hierarchical clustering for better visualization. Black squares were manually added to highlight selected signature clusters.

### Differential cell abundance

Changes in cell type abundance between $Pdgfb^{ret/+}$ and $Pdgfb^{ret/ret}$ derived cells were calculated for every cluster as:

$$\left(\% Pdgfb^{ret/+} \text{cells} - \% Pdgfb^{ret/ret} \text{cells}\right)/\% Pdgfb^{ret/+} \text{cells}$$

We also used MiloR (v1.1)[134] to estimate changes in cell type abundance. Instead of using discrete clusters, MiloR tests for differential abundance in neighborhoods derived from a k-nearest neighbor graph. The parameters used were prop = 0.05, $k$ = 30, $d$ = 20, alpha = 0.2.

### Overrepresentation analyses

GO overrepresentation analysis was performed using Metascape (version 3.5.20240101)[46]. The analysis was done to identify enriched pathways within the same cluster between the $Pdgfb^{ret/+}$ and $Pdgfb^{ret/ret}$ mice, using up- and down-regulated DEGs. All genes with adjusted $p$ value < 0.05 were included for this analysis. DEG lists per cluster were input as a multi-list.

### Human glioma single-cell RNA data and spatial transcriptomics analyses

The human single-cell RNA-seq dataset[24] comprising 18 glioma samples was downloaded from Broad Institute's Single Cell Portal (https://singlecell.broadinstitute.org/single_cell). The dataset was then processed using the basic pipeline of Seurat. The cell-type annotations and UMAP coordinates were kindly provided by the authors. Clusters and results from gene expression level analyses were visualized using the scCustomize package[135].

The previously published human GBM spatial transcriptomic dataset[86] was downloaded from https://datadryad.org/stash/dataset/doi:10.5061/dryad.h70rxwdmj. Data (tumor #UKF 269_T) normalization using SCTransform[136], principal component analysis, and further processing were all done in Seurat using default parameters. Seurat´s AddModuleScore function was used to calculate gene set module scores. The module scores were visualized on the Visium tissue section using Seurat's SpatialFeaturePlot function. The module scores were also aggregated by the different histological areas, which had been annotated applying the Ivy-GAP database classification system[86], of the sample and visualized in heatmaps using the do_EnrichmentHeatmap function as provided in the SCpubr[132] package.

## Cell-cell communication analyses

Cell-cell communication strength is modeled as the probability of ligand-receptor interaction based on the scRNA-seq gene expression data of cells, in conjunction with a database containing prior knowledge of ligands, receptors, and cofactors. To analyze the communication routes between the cell populations in the human glioma dataset (downsampled to 50,000 cells), CellphoneDB (v2.0.0)[27] was used to characterize the number of interactions, and CellChat (v1.4.0)[26] was used to calculate the interaction number and strength using default parameters. Results were visualized using CellChat, ggpubr (v0.6.0)[137], and InterCellar (v2.0.0)[138]. To identify the conserved and context-specific signaling between $Pdgfb^{ret/+}$ and $Pdgfb^{ret/ret}$ pathways, we performed a comparative analysis using CellChat[26]. Cell-cell communication analyses were performed separately for $Pdgfb^{ret/+}$ and $Pdgfb^{+/+}$ derived cells and then compared.

## Spatial transcriptomics sample preparation

Sample preparation was done according to the protocol "Tissue preparation guide" (CG000240; 10x Genomics). All steps were performed using the 10x Visium Spatial Gene Expression Slide & Reagent kit. Methanol fixation and immunofluorescence staining were done according to protocol CG000312. The optimal tissue optimization time (4 min) was determined in a time course experiment according to protocol CG000238RevE. cDNA synthesis and library generation were done according to protocol CG000239RevD. cDNA synthesis was carried out according to the results obtained from qPCR testing with a KAPA FAST SYBR qPCR system (KAPA Biosystems). The amplified cDNA was processed according to protocol, using the SPRIselect Reagent kit (Beckman Coulter). Quality control of the final libraries was performed on an Agilent Bioanalyzer using the High Sensitivity DNA assay (Agilent Biotechnologies). Libraries were sequenced on NovaSeq 6000 (Illumina) using the S1 100 reagent kit (Illumina) with 1% PhiX, and the following sequencing parameters: Read 1−28 cycles, Read 2−90 cycles, Index 1−10 cycles, Index 2−10 cycles.

## Image pre-processing and sequencing analyses

Visium expression slides were imaged on a Leica DMi8 microscope. Images were recorded using the Leica Application Suite (LAS X) software, and computationally cleared with the THUNDER technology (Leica). Images were then imported into Fiji[122], where brightness and contrast were adjusted separately for every channel. Adjusted images were saved as a single multi-stack composite.TIFF file for every sample separately. These.TIFF images were imported into the Loupe Browser (6.5.0, 10x Genomics) for manual alignment. We also saved a separate.TIFF file containing channel 3 (PE/PIM), where brightness was adjusted until the fiducial frames (FF) became visible. These images were imported into the Loupe browser as an auxiliary for FF alignment. Manual alignment of the FF, and manual tissue selection was performed in Loupe Browser (6.5.0, 10x Genomics) following 10x Genomics instructions and recommendations (online Manual Fiducial Alignment guide for Visium). We used Space Ranger count (v2.0.1) to map and count sequencing reads, using the modified reference genome (mm10) containing the RCAS-PDGF sequence (created for mapping the scRNA-seq reads). The multi-stack composite TIFFs (using the darkimage option) and JSON alignments files were used as input for Space Ranger count.

## Spatial transcriptomics data analysis

We used the filtered feature matrix from Space Ranger as an input for downstream analysis. In short, we removed any spots with 0 counts, fewer than 500 genes detected, mitochondrial content higher than 25%, and percent hemoglobin genes detected higher than 20%. We also filtered out mitochondrial and hemoglobin genes. QC and filtering steps were performed in every sample separately. Data analyses were performed as for the scRNA-seq dataset using Seurat (v4.0.3)[126–129]. Batch integration was performed using the Harmony algorithm (v1.0)[130], specifying sample-id and slide as covariate. The function FindConservedMarkers was used for finding DEG between clusters. Gene expression plots were visualized onto the Visium tissue section using Loupe Browser version 6.5.0. The signature scores were visualized onto the Visium tissue section using Seurat's SpatialFeaturePlot function using the option oob = scales::squish. We used the RCTD[71] method, a maximum-likelihood approach, with "full mode" for estimating cell type proportions on spatial transcriptomics spots. We used our scRNA-seq dataset as a reference dataset to deconvolve every spot in our eight spatial transcriptomics samples. The RCTD output is a weighted proportion (0–1), meaning that the sum of all the cell type proportions equals to 1 in every spot. We used the RCTD proportions and the coordinates of each spot as input to the function plotSpatialScatterpie from the SPOTlight package (v 1.6.3)[139] to create a spatial scatterpie chart.

## Patient cohort analyses

TCGA GBM gene expression profiles and matched clinical information were retrieved from the cBioPortal database[83,84]. The TCGA RNA-seq dataset was used for the expression analysis of the primary and recurrent GBM samples, and for the survival analysis, the TCGA Affymetrix HT HG U133A expression dataset was used. OS and PFI data were used to analyze clinical outcomes. The survival analysis was performed using the Surv and survfit functions in the survival package (v3.5-5)[140], and the ggsurvplot function in survminer package (v0.4.9)[141]. We stratified the patients into two expression groups (low and high) according to the median value of the mean expression of the gene signatures (i.e., HSS, ISS, or the top 25 DEGs of cluster MES II). Kaplan−Meier survival curves and log-rank tests were used to evaluate the performance of the gene signatures. The Wilcoxon test was used to assess differences in the expression of pericyte core signatures between primary and recurrent tumors.

## Statistics and reproducibility

Detailed information about the used chemicals, antibodies, probes, cell lines, and software used in this study can be found in the Supplementary Information file.

The maximum possible number of mice was used. These varied across experiments, given the nature of the mouse model and difficulties breeding this mouse strain (non-Mendelian breeding). No data were excluded or censored from the analyses. The Investigators were not blinded to allocation during experiments and outcome assessment. If not specified in each method section, statistical analyses were performed using GraphPad Prism (v9.0, GraphPad Software). Single-cell and spatial transcriptomics analyses were performed in R (v4.3.3)[142].

## Reporting summary

Further information on research design is available in the Nature Portfolio Reporting Summary linked to this article.

## Data availability

scRNA-seq and Visium spatial transcriptomics raw and processed mouse data generated for this study have been deposited in NCBI's Gene Expression Omnibus[143] and are accessible through GEO Series accession numbers GSE272236 and GSE272237, respectively. All other generated data are provided in the main text, the supplementary information file, or the source data file. The publicly available human GBM scRNA-seq data are available through Broad Institute's Single Cell Portal (https://singlecell.broadinstitute.org/single_cell; study: SCP1985) and https://github.com/parveendabas/GBMatlas. The previously published human GBM stRNA-seq was downloaded from https://datadryad.org/stash/dataset/doi:10.5061/dryad.h70rxwdmj. The Xenium human brain cancer dataset was downloaded from the 10X website https://www.10xgenomics.com/datasets/ffpe-human-brain-cancer-data-with-human-immuno-oncology-profiling-panel-and-custom-add-on-1-standard and visualized in Xenium Explorer (v4.0.0, 10x Genomics). The human patient HGG TMA images are available from The Human Protein Atlas (https://images.proteinatlas.org/18144/43118_A_6_8.jpg and https://images.proteinatlas.org/13531/31458_A_5_5.jpg). Source data are provided with this paper.

## Code availability

The code used for the scRNA-seq and spatial transcriptomics analyses is available at https://github.com/KPLab/Pericytes_GBM.

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

## Acknowledgements

K.P. is the Grosskopf Professor of Molecular Medicine at Lund University. This work was supported by grants from the Fru Berta Kamprad foundation to the L2 Cancer Bridge at Create Health, the Swedish Research Council (2023-03136 to K.P.), the Swedish Cancer Society (24 3812 Pj to K.P. and 22 0536 PT to P.B.), the Swedish Childhood Cancer Society (PR2024-0121 to K.P.), the Knut and Alice Wallenberg foundation (to G.B.J. and K.P.), Swedish State Support for Clinical Research through Region Skåne ALF (to K.P.), the Göran Gustafsson foundation (to K.P.), the Mats Paulsson foundations (L2 Cancer Bridge to K.P.), the Cancera foundation (L2 Cancer Bridge to K.P.). The authors thank the Centre for Translational Genomics (Lund University), and Clinical Genomics Lund (SciLifeLab) for sequencing services; Uppsala Multidisciplinary Center for Advanced Computational Sciences (UPPMAX), the Swedish National Infrastructure for Computing (SNIC), and the National Academic Infrastructure for Supercomputing in Sweden (NAISS) for the provided computing and storage resources. The authors would like to thank Ulrike Nuber for kindly providing the *p53*$^{-/-}$, *H-Ras* over-expressing glioma cell line, Laurent Roybon for kindly providing the CMMP vector, Massimo Squatrito for kindly providing the Ntv-a mice, and Christer Betsholtz, as well as Guillem Genové for kindly providing the *Pdgfb*$^{ret/ret}$ mice. The authors would like to thank Wondossen Sime, David Lindgren, Eliane Cortez, and Christina Möller for helpful technical advice and assistance. Finally, the authors would like to thank all reviewers for their feedback.

## Author contributions

Conceptualization: K.P., S.B., A.P. Methodology: P.B., C.O., J.S., M.B., K.H., M.S.T., B.P., E.C., R.R., E.J., G.B.J., S.B. Software: P.B., J.S. Validation: C.O., J.S., P.B., S.B. Formal analysis: P.B., C.O., J.S., K.H., K.P., S.B. Investigation: P.B., C.O., J.S., K.H., B.P., E.C., S.B. Resources: G.B.J., A.P., K.P. Data curation: P.B., C.O., J.S., S.B., K.P. Writing—original draft: S.B., K.P., P.B., J.S. Writing—review and editing: P.B., C.O., J.S., M.B., K.H., M.S.T., B.P., E.C., R.R., E.J., G.B.J., A.P., K.P., S.B. Visualization: P.B., J.S., C.O., S.B. Supervision: K.P., S.B. Project administration: K.P., S.B. Funding acquisition: K.P.

## Funding

## Competing interests

The authors declare no competing interests.
