## [Transparent Peer Review file · Nature Communications]

Pericytes orchestrate a tumor-restraining microenvironment in glioblastoma

Corresponding Author: Professor Kristian Pietras

Version 0:

Reviewer comments:

Reviewer #1

(Remarks to the Author)

The manuscript # NCOMMS-24-61425 addresses the contribution of pericytes in GBM progression specifically in terms of cell type composition and altered signaling pathways that may be critical to drive tumor progression. The authors use PDGFB-ret/+ and PDGFB-ret/ret mice which have reduced numbers of pericytes to examine features of GBM progression and find that pericyte deficiency decreases survival of mice transplanted with the tumor, and promotes more proliferation and growth/invasion of tumor cells. The authors then perform both single-cell RNA sequencing and spatial transcriptomics to characterize differences in cell composition in relationship to the distinct regions of tumors (e.g. hypoxia, perivascular) and identify HGF (derived from macrophages) - cMet (expressed by tumor cells) as a critical pathway for tumor progression that is amplified in PDGFB-ret/ret mice. Finally, the authors examine the ability of c-Met+ or c-Met- cells to grow in vitro and induce tumor growth in vivo and confirm that c-Met+ cells are highly aggressive in triggering death of mice and inducing tumor growth. Finally, the authors examine GBM tumors from patients and identify that groups that have high expression of HSS and ISS signatures have poorer survival.

Although, several studies have been published regarding the role of pericytes in GBM progression and resistance to anticancer therapies, the study presents a novel role for pericytes in tumor cell composition and aberrant signaling pathways derived from infiltrating macrophages (HGF-cMet signaling). The authors use a variety of elegant tools from genetic ablation of pericytes to single cell and spatial transcriptomics to delineate how pericytes contribute to changes in GBM cell composition and which pathways are altered most significantly under these conditions. The study could be improved in a few areas as outlined below:

Major issues:

1) Throughout the paper, the authors use Podocalyxin as a vessel marker. However, Podocalyxin as a marker of vessel polarity and therefore in PDGFBret/ret mice when there are no pericytes and there is BBB leakage, Podocalyxin is not expressed correctly. Therefore, it will be important to use another vessels marker such as CD31 or Glut-1 to confirm changes in vessel area, length, branches (Figure 4A-B). The authors should quantify the BBB leakage in the tumor area in these mice for the images shown in Figure 4C.

2) In the single cell RNAseq data, the authors discuss in the paper changes in distinct cell populations identified from the transcriptome signatures between PDGFB-ret/+ and PDGFB-ret/ret mice. It is important to illustrate them with a graph. In addition, it is important to show the GO analysis of differences in pathways within the same cell type between the PDGFB-ret/+ and PDGFB-ret/ret mice.

3) The authors discuss and show the data for distinct genes that mark populations of macrophages between PDGFB-ret/+ and PDGFB-ret/ret mice. However, they do not show a complete analysis of the myeloid lineage changes (microglia, macrophages etc). This is important for the paper since it reads like a biased analysis by focusing on specific cell populations that express certain gene signatures (see Figures 4H-I).

4) In the spatial transcriptomic analysis shown in Figure 6D for PDGFB-ret/ret mice, it will be important to compare it to PDGFB-ret/+ mice in terms of differences in cell composition (e.g. #1070).

Minor issues:

- 1) The introduction needs to discuss additional studies related to the role of pericytes in GBM progression and resistance to chemotherapy.
- 2) The figure legends need to describe the number of mice used for each experiment (I assumed that each dot of the graph represented one sample but there is a large variability in the number of dots depending on the experiment). For example why are so many dots in the graph shown in Figure 2E? Did the authors analyze such a large number of mice? In addition, the authors need to describe the statistical test used for the assessment of significance.
- 3) Figure 1 is cut. Please correct this error.
- 4) Figure 2F - It looks like all Podo+ vessels regardless of the size are SMA+ in PDGFBret/+ mice. This cannot be true as pericytes in small caliber vessels such as capillaries are SMA-. Please color code the images differently with SMA+ in green and Podo in red.
- 5) The graph in Figure 2J is confusing. The authors should consider color-coding each mouse with a distinct color.
- 6) The larger magnification image in Figure 4F appears stretched along the X-axis. Please ensure that the IF images are not distorted.
- 7) Figure S5 needs to be broken down in 3 supplementary figures.
- 8) The abstract is too long.

Reviewer #2

(Remarks to the Author)

In this manuscript (NCOMMS-24-61425), Braun, Bolivar, et al. provide evidence that pericytes function to restrain tumor growth in glioblastoma (GBM) models. The authors use human single cell RNA-sequencing data to identify pericytes as a hub of cell-cell communication and then mouse model depletion approaches to show that loss of pericyte results in more aggressive tumors that are more invasive and immune suppressive. Along with these observations, the authors also identify Fos1 and MET signaling to be altered in pericyte-depleted tumors. The authors combine functional studies with a variety of -omics approaches (single cell RNA sequencing, spatial transcriptomics) and there is also some correlation to human patient data. In general, this is a complete story that shows the potential function of pericytes in the GBM tumor microenvironment and highlights the need to additional follow up functional studies. While my enthusiasm for this paper is high, there are some issues (ranging from conceptual to technical to narrative) that need to be addressed prior to publication.

1. This paper needs to be placed into the context of reports where cancer stem cells can generate pericyte-like cells and these cells are drivers of malignancy and can be targeted to increase drug penetration into the brain (PMIDs 23540695, 29100012). The results here are opposite of the published papers, which can be for a variety of reasons (models, etc.), and need to be addressed in the introduction and discussion.
2. For the human patient survival data, the changes provided are minimal at best (maybe statistically significant but not biologically significant). I would urge caution to not over interpret these data and in fact, I would move them to the supplement as they are somewhat distracting in the main figures.
3. It is nice to see flow cytometry being used for validation but it would be worth expanding the panels to include other major immune cell types (T cells, etc.).
4. The paper would benefit from a graphical abstract/summary figure.
5. In the abstract, there are several overstatements/overselling that should be adjusted. Remove "unexpectedly" as this pericytes were identified as a top target and in some ways, this makes sense. Remove "extreme" as I am not sure what this biologically means in terms of the mesenchymal phenotype. I would also remove "superior" in terms of invasive capacity ("increased" is fine).
6. In terms of Figure 2, it would be worth being more clear about the survival curves in terms of the models used (indicate directly on the figure).
7. In terms of Figure 3 for the RNA-sequencing, it would be worth being more clear about which model was used in the narrative.
8. In the description of Figure 6, it would likely help the reader to link things more clearly back to changes in pericytes as this section is heavily focused on hypoxia and the mesenchymal phenotype as not so much on pericyte changes.
9. For the methods description of the mice, I would be clear if only 1 sex of both biological sexes were used.

Reviewer #3

(Remarks to the Author)

Braun et al. submitted a manuscript entitled "Pericytes orchestrate a tumor-restraining microenvironment in glioblastoma" for publication in Nature Communications. They aimed to reveal the role of pericytes in glioblastoma (GBM) progression by orchestrating the tumor-suppressive microenvironment. They found that depletion of pericytes promoted mesenchymal tumor cell invasion and tumor-associated macrophages polarization. The topic is interesting; however, the manuscript is very descriptive and without detailed mechanistic studies. Some detailed comments are as follows.

1. It's hard to understand the conclusion for pericytes representing a signaling hub in human GBM, given they only account for about 0.1% of total cells. The interactions between pericytes and other cell types are based on interaction numbers and interaction strength? The interaction is based on ligands (signaling senders) and receptors (recipient)? The conclusion for Fig 1 is very weak, without any validation. Some parts of Fig. 1B were cut.
2. The result subsection: "Human GBM heterogeneity is conserved in a Pdgfb-driven, p53^{-/-} murine high-grade glioma model" is not highly relevant. May move them to supplemental or combine with other subsections.
3. Fig. 2F: the colors of PODXL and α -SMA are very close, making it difficult to distinguish between each other. It's better to change one color into another channel.
4. Fig. 2I: the authors stated that Pdgfbret/ret mice to exhibit a higher luciferase activity than the control animals at all time points. Only one mouse/group and one time point were shown.
5. Does RCAS virus-mediated induction of oncogenic PDGFB expression affect the biology of pericytes?
6. Fig. S2A: the left image used to analyze the hypoxia seems not from the tumor area.
7. Fig. 2: the authors demonstrated that pericyte-deprivation leads to an accelerated tumor progression and premature death in two glioma mouse models. Did they test if pericytes can inhibit tumor cell proliferation in vitro? or they exhibit this effect indirectly?
8. Fig. 3: the authors annotated 4 clusters of cells. What's the difference between cluster 2 (tumor cell-enriched group) and cluster 3 (dividing cells, mostly malignant).
9. Fig. 4C: it is better to stain blood vessel markers with labeled dextrans to show the loss of vascular integrity.
10. The authors mentioned two different invasion modes of OLIG2⁺ tumor cells, by individual cells or by co-option with vessels (however, no evidence is provided).
11. Fig. 4F: why the authors did not unbiased analyzed their single cell seq data to identify the affected TME components, but just directly focused on macrophages?
12. Fig. 4I: the interaction between pericytes and macrophages seems to be no difference between Pdgfbret/+ and Pdgfbret/ret mice, even though Pdgfbret/ret mice have lower numbers of pericytes. Does the impact of pericyte depletion on macrophage polarization attribute to increased endothelial cells? The mechanism is not clear here, and functional studies are needed.
13. The authors demonstrated that three signaling axes known to induce macrophage polarization were exclusively active in Pdgfbret/ret endothelial cells that signaled towards macrophages. However, from their in vitro data showing that only pericyte supernatant, not endothelial cell supernatant, decreases IL4-related macrophage polarization. How to explain these inconsistent results? In addition to polarization, how about macrophage infiltration?
14. Fig. 5A: it is better to highlight the genes that the authors wanted to emphasize, which can make it easier for readers to follow.
15. The legend of Fig. S5D should be upper and bottom panels, rather than left and right.
16. Did the authors test the distribution of pericyte signatures from Fig. 3A in the spatial transcriptomics map?
17. Fig. 6A: the authors showed that MES II signature was more pronounced in Pdgfbret/ret mice with the absence of pericytes. Did they compare the number difference using the scRNA-seq data from Fig. 3A?
18. The authors stated that "Interestingly, we found the MES II top marker Fos1 to be expressed both in hypoxic and tumor peripheral zones." Where is the data?
19. It is not clear why the authors selected the HGF/MET signaling pathway to do the CellChat analysis.
20. Fig. 7D: CD31 or PODXL? The authors mentioned CD31 in the manuscript, but it is shown as PODXL in the figure.
21. MET⁺ tumor cells are more aggressive in vivo. However, the in vitro data from Fig. 7E showed an opposite result. Further studies are needed to dissert the molecular mechanisms underlying this phenomenon.

Reviewer #4

(Remarks to the Author)

Version 1:

Reviewer comments:

Reviewer #1

(Remarks to the Author)

In the revised manuscript # NCOMMS-24-61425A, the authors have addressed several major concerns that were raised by the reviewers in the original submission related to inclusion of additional experiments, images and analyses as well as improving the writing of the paper. There are still some concerns that remain in the revised manuscripts as follows:

1) Concern related to quantification of vessel numbers using Podocalyxin which is a vascular polarity marker (Rev # 1 and #3).

To address this concern, the authors now show staining for CD31 in Figure 4C. I assume that Ret/+ is the image on the left and Ret/Ret is the image on the right (these images are not labelled and need to be labelled in the paper). From the IF staining, it looks like the area covered by CD31+ vessels is smaller in the mutant than heterozygous mice. However, the quantification shown in Fig S4C show the opposite phenotype. How do the authors reconcile these data? Please show representative images for the quantification. Please label the Y axis in Figure S4C as "Total CD31+ vessel area" or "Total CD31+ vessel length" or "Total CD31 Vessel Junctions". Please describe better how the quantification of vessel junctions was performed in the study.

2) Concern related to assessment of vessel leakage (Rev # 1 and #3)

The authors use Dextran 70 kDa to assess the vessel leakage (this is not described in the Methods or Figure legends, just the Result section). However, the method used for analysis of tracer leakage is very odd. The authors quantify "point of tracer leakage" rather than "area of tracer leakage". Why? It looks to me from the images that there is no difference in tracer leakage (there is no tracer leakage) between the tumors in ret/+ or ret/ret. The authors really need to spend some time to repeat this experiment as the data shown does not match the description of the results.

3) Statistics.

The authors state that they used an unpaired t-Test throughout the paper. This analysis will not work for Figure 4J or to test differences in Figures 5G and 5H. What are the tests used in these figures? It would benefit the authors to make the paper clear.

Reviewer #2

(Remarks to the Author)

The authors have been responsive to my previously raised comments. I have no further issues and I think this paper will make a good addition to the literature.

Justin D. Lathia, Cleveland Clinic

Reviewer #3

(Remarks to the Author)

I appreciate the authors' efforts in addressing some of the comments. However, several key concerns remain unresolved, including unexplained inconsistencies in the results or unconvincing data/explanations. Specifically, issues #1, #2, #4, #5, #6, #7, #10, #12, #13, and #21 have not been adequately addressed.

Reviewer #4

(Remarks to the Author)

Version 2:

Reviewer comments:

Reviewer #1

(Remarks to the Author)

The authors have addressed the concerns that I had with the revised submission of the manuscript # NCOMMS-24-61425B. I have no further concerns for the manuscript.

Reviewer #3

(Remarks to the Author)

Thanks the authors for their efforts. I do not have more comments.

Reviewer #4

(Remarks to the Author)

I co-reviewed this manuscript with one of the reviewers who provided the listed reports. This is part of the Nature

Communications initiative to facilitate training in peer review and to provide appropriate recognition for Early Career Researchers who co-review manuscripts.

Reviewer #1 (Remarks to the Author): Expert in scRNA-seq, spatial transcriptomics and pericytes

The manuscript # NCOMMS-24-61425 addresses the contribution of pericytes in GBM progression specifically in terms of cell type composition and altered signaling pathways that may be critical to drive tumor progression. The authors use PDGFB-ret/+ and PDGFB-ret/ret mice which have reduced numbers of pericytes to examine features of GBM progression and find that pericyte deficiency decreases survival of mice transplanted with the tumor, and promotes more proliferation and growth/invasion of tumor cells. The authors then perform both single-cell RNA sequencing and spatial transcriptomics to characterize differences in cell composition in relationship to the distinct regions of tumors (e.g. hypoxia, perivascular) and identify HGF (derived from macrophages) - cMet (expressed by tumor cells) as a critical pathway for tumor progression that is amplified in PDGFB-ret/ret mice. Finally, the authors examine the ability of c-Met⁺ or c-Met⁻ cells to grow in vitro and induce tumor growth in vivo and confirm that c-Met⁺ cells are highly aggressive in triggering death of mice and inducing tumor growth. Finally, the authors examine GBM tumors from patients and identify that groups that have high expression of HSS and ISS signatures have poorer survival.

Although, several studies have been published regarding the role of pericytes in GBM progression and resistance to anticancer therapies, the study presents a novel role for pericytes in tumor cell composition and aberrant signaling pathways derived from infiltrating macrophages (HGF-cMet signaling). The authors use a variety of elegant tools from genetic ablation of pericytes to single cell and spatial transcriptomics to delineate how pericytes contribute to changes in GBM cell composition and which pathways are altered most significantly under these conditions. The study could be improved in a few areas as outlined below:

Major issues:

1) Throughout the paper, the authors use Podocalyxin as a vessel marker. However, Podocalyxin as a marker of vessel polarity and therefore in PDGFBret/ret mice when there are no pericytes and there is BBB leakage, Podocalyxin is not expressed correctly. Therefore, it will be important to use another vessels marker such as CD31 or Glut-1 to confirm changes in vessel area, length, branches (Figure 4A-B). The authors should quantify the BBB leakage in the tumor area in these mice for the images shown in Figure 4C.

Following the reviewer's suggestion we have now performed a re-analysis of the vessel parameters based on immunostaining for CD31. We observed similar changes in vessel area, length and branching, as the ones that were observed based on PODXL quantification. These data are presented on page 7, paragraph 3 in the Results section of the revised manuscript, and have been included in Figure S4C in the revised manuscript. Also, the blood vessel functional analysis with dextran has now been quantified, and it is shown as an extra panel in Figure 4C.

2) In the single cell RNAseq data, the authors discuss in the paper changes in distinct cell populations identified from the transcriptome signatures between PDGFB-ret/+ and PDGFB-ret/ret mice. It is important to illustrate them with a graph. In addition, it is important to show the GO analysis of differences in pathways within the same cell type between the PDGFB-ret/+ and PDGFB-ret/ret mice.

We agree that showing the results of the differential gene expression analyses between PDGF-B^{ret/+} and PDGF-B^{ret/ret} for each cluster separately is important. The complete results are shown in table S4, but we have now also added a multipaneled plot as supplementary figure S5A (one panel for each cluster, that highlights the top DEG between conditions per cluster).

Following the reviewer's suggestion we have also now performed gene ontology overrepresentation analyses using Metascape, and the results are discussed in the last section of page 8 in the Results section and shown in supplementary Figure S6A in the revised manuscript. Metascape analyses was done for all genes together (up and down regulated). Of note, a large number of GO terms related to immune cell processes are differentially regulated between Pdgfb^{ret/+} and Pdgfb^{ret/ret}, corroborating our findings on pericyte deficiency impacting on the immune cell landscape.

3) The authors discuss and show the data for distinct genes that mark populations of macrophages between PDGFB-ret/+ and PDGFB-ret/ret mice. However, they do not show a complete analysis of the myeloid lineage changes (microglia, macrophages etc). This is important for the paper since it reads like a biased analysis by focusing on specific cell populations that express certain gene signatures (see Figures 4H-I).

To answer the reviewer's concern, we have plotted the expression of signature gene lists for 9 macrophage populations in human glioma, described by Abdelfattah et al. (2022, DOI [10.1038/s41467-022-28372-y](https://doi.org/10.1038/s41467-022-28372-y)). These lists include an extended selection of immuno-modulatory genes expressed by myeloid cells. The dotplot included for the referee's perusal (Figure R1A) shows the average expression per cluster in Pdgfb^{ret/+} and Pdgfb^{ret/ret} cells in our dataset. We found that these genes are not altered in the pericyte-poor model, indicating that it is the prototypical polarization markers that we show in Figure 4H that are mostly affected. We originally did not show this figure because the expression of these genes is not as strongly affected by the Pdgfb^{ret/+} and Pdgfb^{ret/ret} condition. Moreover, the gene ontology analysis of differentially expressed genes shown in Figure S6 demonstrated a more comprehensive list of functions that were regulated in macrophages, compared to microglia, including a more pronounced regulation of immune cell functions. Taken together, after careful consideration of the observed gene expression changes in myeloid cells within tumors from control vs pericyte-poor mice, we thus focused on macrophages. These considerations have now been included on pages 8-9 in the Results section of the revised manuscript.

4) In the spatial transcriptomic analysis shown in Figure 6D for PDGFB-ret/ret mice, it will be important to compare it to PDGFB-ret/+ mice in terms of differences in cell composition (e.g. #1070).

Following the reviewer's suggestion, we have now added a similar plot for tumor #1070 from a $Pdgfb^{ret/+}$ mouse in Figure 6D, and the remaining tumors included in the spatial transcriptomics analysis in Figure S12A of the revised manuscript.

Minor issues:

1) The introduction needs to discuss additional studies related to the role of pericytes in GBM progression and resistance to chemotherapy.

The text in the Introduction has been edited following the reviewer's suggestion. Modifications have been done in the third paragraph of the introduction on pages 3-4.

2) The figure legends need to describe the number of mice used for each experiment (I assumed that each dot of the graph represented one sample but there is a large variability in the number of dots depending on the experiment). For example, why are so many dots in the graph shown in Figure 2E? Did the authors analyse such a large number of mice? In addition, the authors need to describe the statistical test used for the assessment of significance.

We apologize for the lack of this information in the original manuscript. We have now added the number of mice used for each experiment/analysis in the figure legend in the revised manuscript. Specifically, the procedure to determine the pericyte vessel coverage is now better described in the 'Experimental Procedures' section. Figure legend 2 has now been edited to clarify that each dot represents a tumor section taken from a distinct tumor position, and the number of mice used for the analysis has been included. We measured tissue sections taken from different tumor areas to reflect the large intratumoral heterogeneity of this glioma model. The applied statistical test has been added to the 'Statistical analyses' section on page 32-33.

3) Figure 1 is cut. Please correct this error.

All figures should be complete in the revised manuscript.

4) Figure 2F - It looks like all Podo+ vessels regardless of the size are SMA+ in PDGRBret/+ mice. This cannot be true as pericytes in small caliber vessels such as capillaries are SMA-. Please color code the images differently with SMA+ in green and Podo in red.

We have changed the figure panels in Figure 2F for images that are displaying a wider spectrum of large, medium-sized and small brain tumor blood vessels. The color coding has been adjusted in a way that the different channels can be distinguished clearly. Indeed, we identified numerous smaller blood vessels that did not exhibit α -SMA staining.

5) The graph in Figure 2J is confusing. The authors should consider color-coding each mouse with a distinct color.

To better illustrate the growth kinetics of each mouse included in the experiment, we have added individual curves in Figure S2C in the revised manuscript.

6) The larger magnification image in Figure 4F appears stretched along the X-axis. Please ensure that the IF images are not distorted.

Width and height proportions were constrained for all panels in Figure 4F in the original manuscript.

7) Figure S5 needs to be broken down in 3 supplementary figures.

As suggested by the reviewers, to provide more clarity, this figure has been distributed across Figure S8, S9, S10 and S11 in the revised manuscript.

8) The abstract is too long.

The abstract has now been shortened.

Reviewer #2 (Remarks to the Author): Expert in GBM, GBM mouse models, and pericytes

In this manuscript (NCOMMS-24-61425), Braun, Bolivar, et al. provide evidence that pericytes function to restrain tumor growth in glioblastoma (GBM) models. The authors use human single cell RNA-sequencing data to identify pericytes as a hub of cell-cell communication and then mouse model depletion approaches to show that loss of pericyte results in more aggressive tumors that are more invasive and immune suppressive. Along with these observations, the authors also identify Fos1 and MET signaling to be altered in pericyte-depleted tumors. The authors combine functional studies with a variety of -omics approaches (single cell RNA sequencing, spatial transcriptomics) and there is also some correlation to human patient data. In general, this is a complete story that shows the potential function of pericytes in the GBM tumor microenvironment and highlights the need to additional follow up functional studies. While my enthusiasm for this paper is high, there are some issues (ranging from conceptual to technical to narrative) that need to be addressed prior to publication.

1. This paper needs to be placed into the context of reports where cancer stem cells can generate pericyte-like cells and these cells are drivers of malignancy and can be targeted to increase drug penetration into the brain (PMIDs 23540695, 29100012). The results here are opposite of the published papers, which can be for a variety of reasons (models, etc.), and need to be addressed in the introduction and discussion.

We have extended the description of the context of our work in the Introduction and Discussion sections of the revised manuscript, referencing the publications suggested by the reviewer, as well as additional papers. Specifically, we now discuss the following papers in the third paragraph of the Introduction (pages 2-3) and on page 19, paragraph 2 in the Discussion section of the revised manuscript:

- Cheng et al (<https://doi.org/10.1016/j.cell.2013.02.021>), detailing the existence of glioma stem cell-derived pericytes.
- Zhou et al (<https://doi.org/10.1016/j.stem.2017.10.002>), detailing targeting opportunities for glioma stem cell-derived pericytes.
- Caspani et al (<https://doi.org/10.1371/journal.pone.0101402>), detailing the physical relationship between pericytes and GBM cells.

2. For the human patient survival data, the changes provided are minimal at best (maybe statistically significant but not biologically significant). I would urge caution to not over interpret these data and in fact, I would move them to the supplement as they are somewhat distracting in the main figures.

As suggested by the reviewer, we have moved the survival analyses to the supplementary information as Figure S14 in the revised manuscript.

3. It is nice to see flow cytometry being used for validation, but it would be worth expanding the panels to include other major immune cell types (T cells, etc.).

We thank the referee for this suggestion. We have now performed FACS analysis of dissociated $Pdgfb^{ret/ret}$ and $Pdgfb^{ret/+}$ gliomas, using antibody panels that include relevant markers to quantify both lymphoid and myeloid immune cell types. Apart from a statistically significant increase of the number of $CD45^+$ immune cells under pericyte-poor conditions, we observe no other major shifts in the abundance of specific immune cell types. However, we note a trend for an increase in $CD11b^+$ myeloid cells and a decrease in $CD45^+$, $CD11b^+$, $LY6C^+$, $CD64^+$, $MHCII^+$ dendritic cells in the absence of pericytes, again consistent with a skewing of the myeloid population towards immune-suppressive functions. The data are now presented on page 8, paragraph 2 in the Results section, and depicted in Figure S4E in the revised manuscript.

4. The paper would benefit from a graphical abstract/summary figure.

The graphical abstract was supplied in the original submission, and we do not know why the reviewers could not view it. We include it now again with the figures.

5. In the abstract, there are several overstatements/overselling that should be adjusted. Remove “unexpectedly” as this pericytes were identified as a top target and in some ways, this makes sense. Remove “extreme” as I am not sure what this biologically means in terms of the mesenchymal phenotype. I would also remove “superior” in terms of invasive capacity (“increased” is fine).

The Abstract has been edited as suggested.

6. In terms of Figure 2, it would be worth being more clear about the survival curves in terms of the models used (indicate directly on the figure).

Figure 2 has now figure titles for each panel that indicate the models used.

7. In terms of Figure 3 for the RNA-sequencing, it would be worth being more clear about which model was used in the narrative.

We have now edited the Results part to clarify that the model used for the scRNA sequencing was RCAS-virus mediated PDGFB-expression and p53 knockdown in Nestin⁺ cells, see Results section page 6, paragraph 1 in the revised manuscript.

8. In the description of Figure 6, it would likely help the reader to link things more clearly back to changes in pericytes as this section is heavily focused on hypoxia and the mesenchymal phenotype as not so much on pericyte changes.

We have now added a clearer summary statement in the last paragraph of this section to link the findings to pericytes, see last paragraph of page 12 in the revised manuscript.

9. For the methods description of the mice, I would be clear if only 1 sex of both biological sexes were used.

We used equal number of male and female mice in all experiments. This has now been clarified in the Methods section "Generation of murine brain tumors", see page 25, of the revised manuscript.

Reviewer #3 (Remarks to the Author): Expert in GBM, TME and macrophage Braun et al. submitted a manuscript entitled “Pericytes orchestrate a tumor-restraining microenvironment in glioblastoma” for publication in Nature Communications. They aimed to reveal the role of pericytes in glioblastoma (GBM) progression by orchestrating the tumor-suppressive microenvironment. They found that depletion of pericytes promoted mesenchymal tumor cell invasion and tumor-associated macrophages polarization. The topic is interesting; however, the manuscript is very descriptive and without detailed mechanistic studies. Some detailed comments are as follows.

1. It's hard to understand the conclusion for pericytes representing a signaling hub in human GBM, given they only account for about 0.1% of total cells. The interactions between pericytes and other cell types are based on interaction numbers and interaction strength? The interaction is based on ligands (signaling senders) and receptors (recipient)? The conclusion for Fig 1 is very weak, without any validation. Some parts of Fig. 1B were cut.

The interactions between cell types are estimated as a function of both the number of interactions and the interaction strength. The interaction strength is estimated based on ligand-receptor mediated communication probability between cell types (both sender and receiver), based on the mean gene expression levels per cluster. Significant interactions are assessed by permutation tests. By default, CellChat does not account for the effect of cell type proportions in the probability calculation. We estimated the communication probability or strength using the function computeCommunProb() with the parameter population.size=FALSE (default), that computes the probability of interaction based only on the expression levels of ligands and receptors but not the population size. Setting population.size=TRUE would instead account for the abundance of cells, in which case we would expect large clusters to show more and stronger interactions than small clusters just because of their size. For this study, we used the default parameter precisely because population size differences are overwhelming. When population size is not considered, pericytes indeed have a disproportionately high number of interactions and overall interaction strength, especially when looking at outgoing interactions. This pattern is observed in the human glioblastoma dataset (Figure 1 C-D and Figure S1 B-C) and validated in the mouse model analyzed in this study (Figure S4A and S4B), i.e. in two distinct datasets from two different species.

Notably, we believe that pericytes are underrepresented in scRNAseq datasets in general, as incomplete tissue dissociation, pericyte morphology, and limitations to use harsher enzymatic conditions, as well as library preparation may affect this cell type in a disproportionate manner in our experience. IF analysis of murine glioma sections shows that cells are more abundant in situ than accounted for in scRNAseq data (Figure R1B). We believe that it is valuable to look at cellular communication based on the average expression levels of ligands and receptor pairs per cluster and not based on the cluster sizes in order to find the qualitatively most interesting biological pathways that instruct the community about the mechanistic underpinnings of the glioma microenvironment.

Figure 1B has been corrected.

2. The result subsection: “Human GBM heterogeneity is conserved in a Pdgfb-driven, p53^{-/-} murine high-grade glioma model” is not highly relevant. May move them to supplemental or combine with other subsections.

We have moved, abbreviated and removed parts of the mentioned paragraph, as suggested by the reviewer. However, we feel it would be important to keep parts of this paragraph in the Results section. A detailed description of the annotated cell compartments, its heterogeneity and baseline character is crucial for the reader to be able to comprehend the shifts that occur to the tumor ecosystem upon pericyte deprivation. Naturally, should the referee and editors insist that the information be moved to other sections, or completely omitted, we are open to re-evaluating this stance.

3. Fig. 2F: the colors of PODXL and α -SMA are very close, making it difficult to distinguish between each other. It's better to change one color into another channel.

We have now modified the colors of Fig. 2F in the revised manuscript.

4. Fig. 2I: the authors stated that Pdgfb^{ret/ret} mice to exhibit a higher luciferase activity than the control animals at all time points. Only one mouse/group and one time point were shown.

The average growth curve of mice from each genotype based on the luminescence read-out is shown in Figure 2J, and the growth curve of each individual mouse is included in Figure S2C in the revised manuscript.

5. Does RCAS virus-mediated induction of oncogenic PDGFB expression affect the biology of pericytes?

We thank the reviewer for this relevant question. Abramsson et al. (2003, DOI [10.1172/JCI18549](https://doi.org/10.1172/JCI18549)) have shown that an endothelial source of PDGF-B appears to be required for the establishment of a tight association between the endothelial and mural cells in tumor vessels, and that this source cannot be compensated for by tumor-derived PDGF-B. They transplanted fibrosarcoma T241 cells, overexpressing PDGF-B, in Pdgfb^{ret/ret} mice, and observed a partial or complete detachment of the majority of pericytes from the vessel wall, irrespective of whether the T241 cells expressed PDGF-B or not, indicating that the proper arrangement of pericytes in the vessel wall was dependent on the expression of PDGF-B by the host endothelium (Abramsson et al., 2003). Importantly, our analysis of the pericyte coverage of RCAS-induced tumors in Pdgfb^{ret/ret} mice demonstrate that there is a robust reduction in pericyte investment (Figure 2E and 2F in the revised manuscript), comparable to that observed in the other tumor models that are not relying on oncogenic PDGF-B.

6. Fig. S2A: the left image used to analyze the hypoxia seems not from the tumor area.

The center of the imaged tumor section features a zone of very high nuclear density and pseudopalisading necrosis which represents the tumor core. However, virtually the complete tissue section shows an abnormally high cellular density and can be considered either tumor tissue or highly infiltrated tumor invasive zone, as indicated by the prolific hypoxic areas.

7. Fig. 2: the authors demonstrated that pericyte-deprivation leads to an accelerated tumor progression and premature death in two glioma mouse models. Did they test if pericytes can inhibit tumor cell proliferation in vitro? or they exhibit this effect indirectly?

Thank you for raising this concern. To address this, we have now tested the effect of pericytes on tumor cell proliferation in vitro. Using a transwell-based co-culture approach, we cultured Pdgf-b, p53 kd tumor spheres with pericytes or endothelial cells over 5 days. We subsequently performed a BrdU/To-Pro-3-based FACS analysis, that showed a mild proliferation-increasing effect on glioma cells for both pericytes and endothelial cells, as compared to the control group (Figure R1C). Whereas these results are in line with literature referenced in our manuscript, and underline the complex character of pericyte-tumor cell interactions, they do not question the conclusions of our work, as the model we present in our study is not based on direct interactions, but on the indirect effects of TME remodeling observed in connection with pericyte deprivation. Given the comparably small effects observed on tumor cell proliferation in vitro, and lack of evidence for a growth reduction of tumor cells in vivo in the absence of pericytes, we have decided not to include these data in the revised manuscript.

8. Fig. 3: the authors annotated 4 clusters of cells. What's the difference between cluster 2 (tumor cell-enriched group) and cluster 3 (dividing cells, mostly malignant).

These two clusters differ in the cell cycle scores and further classification into cell cycle states (Figure R1D). We estimated cell cycle scores per cell using Seurat. The list of genes used in cell-cycle regression (already defined in Seurat), is divided into two sublists: 1) genes associated with S-phase and 2) genes associated with the G2M-phase. This analysis suggests that cells in cluster 2 are (mostly) in the G1 phase, while cells within cluster 3 are in the S or G2M phase.

9. Fig. 4C: it is better to stain blood vessel markers with labeled dextrans to show the loss of vascular integrity.

As suggested by the reviewer, we have analyzed vascular integrity by perfusion with labeled dextrans, indicating leakage of vascular fluid into the perivascular space upon pericyte deprivation (Figure 4C, left panels). In addition, we have now quantified vessel leakage (Figure 4C, right panel), demonstrating the loss of patency of the BBB in the pericyte-poor state.

10. The authors mentioned two different invasion modes of OLIG2⁺ tumor cells, by individual cells or by co-option with vessels (however, no evidence is provided).

Using a mouse model that allows for variation in Olig2 functional status (Olig2^{cre/+}; Trp53^{fl/fl}; hEGFR^{vIII}), Griveau et al. analyzed vascular relationships of Olig2⁺ versus Olig2⁻ gliomas (Griveau et al., 2018, DOI [10.1016/j.ccell.2018.03.020](https://doi.org/10.1016/j.ccell.2018.03.020)), using, among other methods, IF analysis. They observed that OPC cells of Olig2⁺ tumors preferentially invade following tumor capillary tracks, i.e. via single-cell vessel co-option, while Olig2⁻ invading tumor cells tend to invade physiologic brain parenchyma as dense perivascular collections. Thus, the authors found different modes of invasion depending on the glioma cell state. In the mouse model used for this study, which is mechanistically different, we quantified tumor cells of the invasive zone based on IF analysis of Olig2⁺ GBM cells (Figures 4D, 4E and S4D), and found increased proportions of co-opting cells in the absence of pericytes. We interpreted these results as consequential to a remodeled perivascular space and possibly altered transcriptional programs of infiltrating tumor cells in pericyte-deprived gliomas, allowing for a more effective mode of glioma cell infiltration of the surrounding brain parenchyma. These findings are especially interesting in light of recently published data stating an increased glioma cell invasion in recurring IDH-wildtype gliomas, known to be pericyte-deprived, that was linked to an altered tumor microenvironment (Varn et al., 2022, DOI [10.1016/j.cell.2022.04.038](https://doi.org/10.1016/j.cell.2022.04.038)).

11. Fig. 4F: why the authors did not unbiased analyzed their single cell seq data to identify the affected TME components, but just directly focused on macrophages?

After annotating the UMAP clusters of the scRNA-seq data set, we identified the top differentially expressed genes between Pdgfb^{ret/ret} vs Pdgfb^{ret/+} for each cluster (Table S4 and Figure S5A). The most striking alterations in terms of signature changes were identified for the cluster representing macrophages, indicating an increase in expression for macrophage polarization markers, and suggesting a shift of certain myeloid cell populations toward a tumor-supporting state upon pericyte deprivation. The notion of a changing functional profile was corroborated by a GO analysis of our scRNA-seq dataset that we have now added to the revised manuscript (Figure S6A): the GO analysis shows, among other alterations, a changed cytokine signaling pattern of tumor macrophages in pericyte-poor conditions (see also reviewer 1, question 2). Thus, our initial scrutiny of the full dataset led us to focus our efforts on macrophages that exhibited an interesting change of phenotype upon pericyte deprivation.

12. Fig. 4I: the interaction between pericytes and macrophages seems to be no difference between *Pdgfbret/+* and *Pdgfbret/ret* mice, even though *Pdgfbret/ret* mice have lower numbers of pericytes. Does the impact of pericyte depletion on macrophage polarization attribute to increased endothelial cells? The mechanism is not clear here, and functional studies are needed.

We thank the referee for allowing us to clarify this point. The literature clearly states pericyte functions of regulating and maintaining the endothelium in the physiologic context (Armulik et al., 2010, DOI [10.1038/nature09522](https://doi.org/10.1038/nature09522); Gaengel et al., 2009, DOI [10.1161/ATVBAHA.107.161521](https://doi.org/10.1161/ATVBAHA.107.161521)), while we and others find a diminished pericyte:endothelial cell ratio to lead to a remodeled and more defective tumor vasculature (Figure 2 A-F, (Hong et al., 2015, DOI [10.1093/jnci/djv209](https://doi.org/10.1093/jnci/djv209)) in terms of morphology, functionality and paracrine signaling. Due to the co-localization of perivascular macrophages and endothelial cells, the upregulation of several cell-cell interaction pathways that affect polarization of macrophages (e.g. CD200, GAS, CSF, TGF β), and an altered signaling of macrophages to glioma cells that we find in this work, we conclude that a remodeled endothelium shifts macrophages toward a tumor-promoting state in pericyte-poor gliomas.

To further prove this point, we have now performed an experiment in which we added the ligand GAS6, which was upregulated by the endothelium in the absence of pericytes, to cultured macrophages and evaluated its influence on macrophage polarization. We found GAS6 to reinforce alternative activation of macrophages in synergy with IL4, and have included these results now in page 9, paragraph 1 and Figure S7B in the revised manuscript.

13. The authors demonstrated that three signaling axes known to induce macrophage polarization were exclusively active in *Pdgfbret/ret* endothelial cells that signaled towards macrophages. However, from their in vitro data showing that only pericyte supernatant, not endothelial cell supernatant, decreases IL4-related macrophage polarization. How to explain these inconsistent results? In addition to polarization, how about macrophage infiltration?

We thank the reviewer for pointing out this apparent inconsistency. It should be emphasized that the assessment of macrophage polarization by endothelial- or pericyte-conditioned medium was performed in vitro, and thus may not fully represent the more complex situation in vivo. Thus, it is not clear whether the phenotype of endothelial cells cultured on stiff plastic would be more similar to the pericyte-rich or -poor situation. Moreover, the IL4-induced alternative activation of macrophages used in the experimental setup yields an almost full polarization, making it difficult to appreciate a synergistic effect with other factors. Taken together, the main point that we would like to make with this experiment is the conceptually novel possibility that pericytes directly impinge on macrophage polarization markers, although more work is needed to mechanistically probe this function.

In our additional immune marker profiling, we found a significant increase in CD45⁺ cells in Pdgfb^{ret/ret} tumors that, considering the relative abundance of all other immune cell populations, can only arise from myeloid cells (Figure S4E). The higher abundance of macrophages and other myeloid cells in Pdgfb^{ret/ret} tumors is corroborated by our scRNA-seq analysis (Table S3) and in line with published data (Hong et al., 2015).

14. Fig. 5A: it is better to highlight the genes that the authors wanted to emphasize, which can make it easier for readers to follow.

Following the reviewer's suggestion, we have reduced the number of genes in the main plot to focus on the genes we want to emphasize (see Figure 5A in the revised manuscript). We have also added an extra supplementary figure that shows the rest of the markers that were present in the original figure (see Figure S8A in the revised manuscript).

15. The legend of Fig. S5D should be upper and bottom panels, rather than left and right.

We thank the reviewer for pointing out this mistake; this figure is now Figure S8E in the revised manuscript, and the figure legend has been corrected.

16. Did the authors test the distribution of pericyte signatures from Fig. 3A in the spatial transcriptomics map?

Following the reviewer's suggestion, we have plotted the mean z-score of the top 50 DEG from the pericyte cluster of our scRNA-seq dataset onto the spatial dataset (see Figure R1E for the referee's perusal). The control sample exhibited a distribution of the pericyte signature consistent with a vascular pattern. As expected, the pericyte signature was more weakly expressed within the map of the pericyte-deprived tumor, but the pattern was still indicative of residual signature expression within the microvasculature.

17. Fig. 6A: the authors showed that MES II signature was more pronounced in Pdgfb^{ret/ret} mice with the absence of pericytes. Did they compare the number difference using the scRNA-seq data from Fig. 3A?

The abundance of glioma cells carrying the MES II signature in our scRNA-seq dataset has been compared, and we refer to that in the manuscript: "Notably, the overall percentage of tumor cells decreased, whereas we observed the opposite trend for MES II cells when comparing Pdgfb^{ret/+} versus Pdgfb^{ret/ret} gliomas (34% increase in Pdgfb^{ret/ret} mice; Table S3), indicating that the loss of pericytes favours the expansion of this extreme mesenchymal glioma cell state."

18. The authors stated that “Interestingly, we found the MES II top marker *Fosl1* to be expressed both in hypoxic and tumor peripheral zones.” Where is the data?

*We thank the reviewer for pointing this out. We have now added Figure 5E to the revised manuscript, showing the expression of *Fosl1* in the spatial transcriptomics samples (exemplified by #1159).*

19. It is not clear why the authors selected the HGF/MET signaling pathway to do the CellChat analysis.

Active HGF-MET signaling is associated with progression of many different tumor types, among them GBM, by reinforcing enrichment of cancer stem cells, tumor cell migration, invasion and angiogenesis (Cheng and Guo, 2019, DOI [10.1186/s13046-019-1269-x](https://doi.org/10.1186/s13046-019-1269-x), Khater and Abou-Antoun, 2021, DOI [10.3389/fcell.2021.654103](https://doi.org/10.3389/fcell.2021.654103), Mulcahy et al., 2020, DOI [10.3390/ijms21207546](https://doi.org/10.3390/ijms21207546)), and serving as a regulator of EMT (Boccaccio and Comoglio, 2013, DOI [10.1158/0008-5472.CAN-12-4039](https://doi.org/10.1158/0008-5472.CAN-12-4039)).

*Importantly, the HGF/MET pathway (and other highlighted pathways in the manuscript) were not selected in advance. This pathway was identified by CellChat as a pathway showing a significant change between the *Pdgfb*^{ret/+} and *Pdgfb*^{ret/ret} conditions. Cellchat showed that communication between *Hgf*-expressing macrophages and receiving *Met*-expressing MES II cells occurs only upon pericyte deprivation (in *Pdgfb*^{ret/ret} samples). Figure 7B shows that this communication does not occur in the *Pdgfb*^{ret/+} condition due to the lack of *Met* expression in MES II cells. Upregulation of *Met*, together with other differentially expressed genes such as the GBM mesenchymal subtype master regulator *Fosl1*, suggests that a strong mesenchymal profile of MES II cells was promoted upon pericyte loss. Furthermore, the association of mesenchymal glioma cells with tumor associated macrophages (TAM) represent a central component of this subtype and have been shown to express HGF (Dong et al., 2019, DOI [10.1038/s41416-019-0482-x](https://doi.org/10.1038/s41416-019-0482-x)). Together with our finding of more abundant and polarized TAM in *Pdgfb*^{ret/ret}, these results led us to perform a cell-cell interaction analysis of the HGF-MET pathway that showed exclusive signaling in *Pdgfb*^{ret/ret} mice, linking together a remodelled TME with a reinforced mesenchymal cell state upon pericyte deprivation. These findings thus mechanistically couple a remodelled TME with a reinforced mesenchymal cell state upon pericyte deprivation.*

20. Fig. 7D: CD31 or PODXL? The authors mentioned CD31 in the manuscript, but it is shown as PODXL in the figure.

For the IF analysis shown in Figure 7D, we immunostained for PODXL, and not for CD31. The error has been corrected, thank you for pointing this out.

21. MET⁺ tumor cells are more aggressive in vivo. However, the in vitro data from Fig. 7E showed an opposite result. Further studies are needed to dissect the molecular mechanisms underlying this phenomenon.

It has been demonstrated that the MET/HGF axis can potentially transform neural stem cells into glioma-initiating stem cells, and reinforce tumor cell invasiveness (Li et al, 2011, DOI [10.1073/pnas.1016912108](https://doi.org/10.1073/pnas.1016912108); Joo 2012, DOI [10.1158/0008-5472.CAN-11-3760](https://doi.org/10.1158/0008-5472.CAN-11-3760)). Although HGF-autocrine glioma cell activation has been shown (Qin 2020, DOI [10.1093/naajnl/vdaa067](https://doi.org/10.1093/naajnl/vdaa067)), our data indicate paracrine signaling with macrophages as the primary source for HGF (Figure 7A and 7B).

In vitro experiments, lacking HGF availability from the microenvironment, hamper the MET/HGF axis, which is reflected in the lower sphere induction potential of MET⁺ compared to MET⁻ control cells that might exploit other growth-promoting pathways. In a new experiment, we have added HGF to MET⁺ and MET⁻ cell cultures in vitro to mimic the microenvironmental supply of this ligand during pericyte-poor conditions, and compared sphere induction capability. In line with our hypothesis, we observed HGF treatment to induce a significant increase in sphere induction potential in MET⁺ as compared to MET⁻ cells. We did not find differences in cell numbers in this experiment, most likely because cell number is not always correlated with sphere induction rate, but changes become obvious during later passages. These data are now discussed in the Results section on page 13-14, and included as Figure 7E and 7F in the revised manuscript.

Reviewer #4 (Remarks to the Author): Early-Career Researcher co-reviewer

Figure R1

Additional experimental procedures (related to figure R1C).

Glioma cells derived from PDGFB-induced, p53 kd murine tumors were grown as spheres. 2×10^4 passaged single glioma cells were cultured in transwells (0.4 μm pore size, Costar) together with 5×10^4 pericytes, endothelial cells or glioma cells (controls) for 5d. Subsequently, the samples were labeled with BrdU for 5h (Thermofisher, #8811-6600-42), stained with To-Pro-3 (ToPro) and analyzed with a BD FACS Melody system. FACS data was analyzed with FlowJo, and tested for statistical significance with t-test.

Legend to figure R1. Additional data.

- (A) Expression of 9 gene lists representing myeloid cell populations taken from Abdelfattah et al. (2023) in the scRNAseq dataset.
- (B) Representative fluorescence images of *Pdgfb*^{ret/+} glioma derived tissue section immune stainings for PODXL and PDGFR β , indicating the abundance of pericytes in the glioma model.
- (C) Transwell assays of *Pdgf-b*, p53 kd tumor spheres, cultured with pericytes or endothelial cells over 5d, and subsequent BrdU/To-PRO-3-based FACS analysis.
- (D) UMAP plot showing cell cycle states of the scRNA-seq dataset.
- (E) Pericyte cluster signature score of the scRNA-seq dataset, plotted onto the spatial dataset.

Reviewer #1 (Remarks to the Author):

In the revised manuscript # NCOMMS-24-61425A, the authors have addressed several major concerns that were raised by the reviewers in the original submission related to inclusion of additional experiments, images and analyses as well as improving the writing of the paper.

Author response, 2nd revision

Thank you for acknowledging our efforts to improve the study according to the insightful comments from the referees.

There are still some concerns that remain in the revised manuscripts as follows:

1) Concern related to quantification of vessel numbers using Podocalyxin which is a vascular polarity marker (Rev # 1 and #3).

To address this concern, the authors now show staining for CD31 in Figure 4C. I assume that ret/+ is the image on the left and ret/ret is the image on the right (these images are not labelled and need to be labelled in the paper). From the IF staining, it looks like the area covered by CD31+ vessels is smaller in the mutant than heterozygous mice. However, the quantification shown in Fig S4C show the opposite phenotype. How do the authors reconcile these data? Please show representative images for the quantification. Please label the Y axis in Figure S4C as "Total CD31+ vessel area" or "Total CD31+ vessel length" or "Total CD31 Vessel Junctions". Please describe better how the quantification of vessel junctions was performed in the study.

Author response, 2nd revision

We apologize for the missing labels, these have now been reintroduced into the images, and we have amended the Y axis label as suggested. Additionally, we have changed the IF images to better reflect all vessel-related phenotypes, as the previous ones were chosen to focus on the vessel leakage. The method for quantification of vessel junctions has now been described in more detail, see pages 25-26 in the revised manuscript.

2) Concern related to assessment of vessel leakage (Rev # 1 and #3)

The authors use Dextran 70 kDa to assess the vessel leakage (this is not described in the Methods or Figure legends, just the Result section). However, the method used for analysis of tracer leakage is very odd. The authors quantify "point of tracer leakage" rather than "area of tracer leakage". Why? It looks to me from the images that there is no difference in tracer leakage (there is no tracer leakage) between the tumors in ret/+ or ret/ret. The authors really need to spend some time to repeat this experiment as the data shown does not match the description of the results.

Author response, 2nd revision

The Methods section and the Figure legend state that Dextran 70 kDa was used to assess vessel leakage. The reason for quantifying points of tracer leakage was to demonstrate a widespread break-down of the blood-brain barrier, rather than an excessive leakage from a few localized vessels. As suggested by the Referee, we have now quantified the area of tracer leakage. In order not to skew the results due to differences in vessel density, the leakage area was expressed relative to the vessel area. Similar to our previous analyses, loss of pericytes induced significant leakage of the Dextran tracer. These data have now replaced the earlier analysis in Figure 4C in the revised manuscript. Also, we apologize for the poor quality of the images of tracer leakage, which have now been replaced with representative pictures in Figure 4C of the revised manuscript.

3) Statistics.

The authors state that they used an unpaired t-Test throughout the paper. This analysis will not work for Figure 4J or to test differences in Figures 5G and 5H. What are the tests used in these figures? It would benefit the authors to make the paper clear.

Author response, 2nd revision

We thank the Referee for pointing out this omission. The FACS-based quantification of CD206+ M ϕ shown in Figure 4J consists of three parallel experiments where different conditions were tested, and that we decided to show in one graph for better clarity. Since the proportion of CD206+ M ϕ exposed to different media within each treatment group (and not the treatment groups *per se*) are compared, we performed unpaired t-tests within each treatment group.

For Figure 5G-H we have used comparisons of Z-scores and RCTD-scores, and their estimations are described in detail under "Experimental Procedures". Z-scores are estimated to assess how many standard deviations from the mean a value is (either above or below) and it thus accounts for the variation present in the dataset. As the Referee correctly points out, a t-test is not applicable for this type of analysis and instead z-scores were directly compared without using a statistical test. The comparison showed that the scores are highest in the hypoxic areas and subsequently decrease with the distance from hypoxia. In analogy, it is inappropriate to use t-test for RCTD scores, that represent proportions of cells, and we used that algorithm to directly compare the co-location of cell types with certain tissue zones.

Reviewer #2 (Remarks to the Author):

The authors have been responsive to my previously raised comments. I have no further issues and I think this paper will make a good addition to the literature.

Justin D. Lathia, Cleveland Clinic

Author response, 2nd revision

Thank you, we appreciate the constructive comments that helped us improve the paper.

Reviewer #3 (Remarks to the Author):

I appreciate the authors' efforts in addressing some of the comments. However, several key concerns remain unresolved, including unexplained inconsistencies in the results or unconvincing data/explanations. Specifically, issues #1, #2, #4, #5, #6, #7, #10, #12, #13, and #21 have not been adequately addressed.

Author response, 2nd revision

Following a dialogue with the Editor, we here offer further data, insights and explanations to clarify the Referee's remaining concerns.

#1 It's hard to understand the conclusion for pericytes representing a signaling hub in human GBM, given they only account for about 0.1% of total cells. The interactions between pericytes and other cell types are based on interaction numbers and interaction strength? The interaction is based on ligands (signaling senders) and receptors (recipient)? The conclusion for Fig 1 is very weak, without any validation. Some parts of Fig. 1B were cut.

Author response, 1st revision

The interactions between cell types are estimated as a function of both the number of interactions and the interaction strength. The interaction strength is estimated based on ligand-receptor mediated communication probability between cell types (both sender and receiver), based on the mean gene expression levels per cluster. Significant interactions are assessed by permutation tests. By default, CellChat does not account for the effect of cell type proportions in the probability calculation. We estimated the communication probability or strength using the function `computeCommunProb()` with the parameter `population.size=FALSE` (default), that computes the probability of interaction based only on the expression levels of ligands and receptors but not the population size. Setting `population.size=TRUE` would instead account for the abundance of cells, in which case we would expect large clusters to show more and stronger interactions than small clusters just because of their size. For this study, we used the default parameter precisely because population size differences are overwhelming. When population size is not considered, pericytes indeed have a disproportionately high number of interactions and overall interaction strength, especially when looking at outgoing interactions. This pattern is observed in the human glioblastoma dataset (Figure 1 C-D and Figure S1 B-C) and validated in the mouse model analyzed in this study (Figure S4A and S4B), i.e. in two distinct datasets from two different species.

Notably, we believe that pericytes are underrepresented in scRNAseq datasets in general, as incomplete tissue dissociation, pericyte morphology, and limitations to use harsher enzymatic conditions, as well as library preparation may affect this cell type in a disproportionate manner in our experience. IF analysis of murine glioma sections shows that cells are more abundant in situ than accounted for in scRNAseq data (Figure R1B). We believe that it is valuable to look at cellular communication based on the average expression levels of ligands and receptor pairs per cluster and not based on the cluster sizes in order to find the qualitatively most interesting biological pathways that instruct the community about the mechanistic underpinnings of the glioma microenvironment.

Figure 1B has been corrected.

Author response, 2nd revision

We understand that the finding that a rare cell type, such as the pericyte, is the most active signaling partner in the GBM microenvironment may be conceived as surprising. However, taking into account the strategic positioning of pericytes, intimately associated with endothelial cells and trafficking immune cells, and open for interactions with the surrounding tumor parenchyma, the intense paracrine engagement by pericytes is more logical. Using a spatial transcriptomics dataset, and corroborated by immunostaining, we now illustrate the tissue organization of human GBM from the pericyte perspective. As demonstrated in Figure 1E and S1D in the revised manuscript, pericytes do hold a central position with the ability to make contact with many, if not all, other constituent cell types of GBM. Taken together, the data presented in Figure 1 serve as the starting point for our further explorations of the functional implications of the paracrine interactions by pericytes that was uncovered by our in-depth analysis of scRNA-seq data.

#2 The result subsection: “Human GBM heterogeneity is conserved in a Pdgfb-driven, p53^{-/-} murine high-grade glioma model” is not highly relevant. May move them to supplemental or combine with other subsections.

Author response, 1st revision

We have moved, abbreviated and removed parts of the mentioned paragraph, as suggested by the reviewer. However, we feel it would be important to keep parts of this paragraph in the Results section. A detailed description of the annotated cell compartments, its heterogeneity and baseline character is crucial for the reader to be able to comprehend the shifts that occur to the tumor ecosystem upon pericyte deprivation. Naturally, should the referee and editors insist that the information be moved to other sections, or completely omitted, we are open to re-evaluating this stance.

Author response, 2nd revision

As the Referees suggested, we have deleted the Results subsection “Human GBM heterogeneity is conserved in a PDGFB-driven, p53^{-/-} murine high-grade glioma model” together with most of the text, and moved the remaining crucial points to other subsections and to the Experimental Procedures section, see page 29 in the revised manuscript.

#4 Fig. 2I: the authors stated that Pdgfb^{ret}/ret mice to exhibit a higher luciferase activity than the control animals at all time points. Only one mouse/group and one time point were shown.

Author response, 1st revision

The average growth curve of mice from each genotype based on the luminescence read-out is shown in Figure 2J, and the growth curve of each individual mouse is included in Figure S2C in the revised manuscript.

Author response, 2nd revision

To further demonstrate the luciferase read-out in GBM developing in Pdgfb^{ret/+} vs Pdgfb^{ret/ret} mice, we have now included images from a representative pair of mice in Supplemental Figure 3B in the revised manuscript.

#5 Does RCAS virus-mediated induction of oncogenic PDGFB expression affect the biology of pericytes?

Author response, 1st revision

We thank the reviewer for this relevant question. Abramsson et al. (2003, DOI [10.1172/JCI18549](https://doi.org/10.1172/JCI18549)) have shown that an endothelial source of PDGF-B appears to be required for the establishment of a tight association between the endothelial and mural cells in tumor vessels, and that this source cannot be compensated for by tumor-derived PDGF-B. They transplanted fibrosarcoma T241 cells, overexpressing PDGF-B, in *Pdgf-b^{ret/ret}* mice, and observed a partial or complete detachment of the majority of pericytes from the vessel wall, irrespective of whether the T241 cells expressed PDGF-B or not, indicating that the proper arrangement of pericytes in the vessel wall was dependent on the expression of PDGF-B by the host endothelium (Abramsson et al., 2003). Importantly, our analysis of the pericyte coverage of RCAS-induced tumors in *Pdgf-b^{ret/ret}* mice demonstrate that there is a robust reduction in pericyte investment (Figure 2E and 2F in the revised manuscript), comparable to that observed in the other tumor models that are not relying on oncogenic PDGF-B.

Author response, 2nd revision

To strengthen the proposition that the phenotype observed in GBM developing under pericyte-poor conditions is independent of oncogenic PDGF-BB, we would like to point out that we have performed experiments in several mouse models of cancer. Transplantation of neurospheres transformed by knock-out of *Trp53* and expression of oncogenic H-Ras into the brain of *Pdgfb^{ret/ret}* mice produces a similar acceleration of tumor progression as induction of GBM that is driven by oncogenic *PDGFB* expression in RCAS-tva mice (Figure 2B). Similarly, GBM formed by orthotopic transplantation of GL261 (also independent of oncogenic PDGF-BB) exhibit the same degree of pericyte loss as *PDGFB*-driven tumors when transplanted into *Pdgfb^{ret/ret}* mice. To further underline these facts, we have now immunostained tumor tissue from the *PDGFB*-dependent RCAS-tva model and from the *PDGFB*-independent transformed neurosphere model for canonical markers of pericytes (PDGFRb and ACTA2 (α -SMA)). As shown in Figure 2A for reviewers, the tissue organization with regard to pericytes and mural cells is very similar, with PDGFRb labeling pericytes surrounding small capillaries and mural cells in large-diameter blood vessels, and ACTA2 (α -SMA) labeling predominantly mural cells in the larger blood vessels.

Taken together, our validation of phenotypes in mouse models with different oncogenic drivers, as well as our previous account of the literature demonstrating convincingly that pericyte biology is not affected by tumor cell-derived PDGF-BB, there is strong evidence for the effects of pericytes uncovered by our studies being universal, independent of the oncogenic driver.

#6 Fig. S2A: the left image used to analyze the hypoxia seems not from the tumor area.

Author response, 1st revision

The center of the imaged tumor section features a zone of very high nuclear density and pseudopalisading necrosis which represents the tumor core. However, virtually the complete tissue section shows an abnormally high cellular density and can be considered either tumor tissue or highly infiltrated tumor invasive zone, as indicated by the prolific hypoxic areas.

Author response, 2nd revision

To further illustrate the tumor boundaries, we have now visualized malignant cells by immunostaining for OLIG2, a well-characterized pathological marker for gliomas, in particular

of the OPC/NPC subtype. We have now replaced the images in Figure S2A in the revised manuscript for new ones in which the tumor:brain boundary is labeled based on OLIG2-positivity.

#7 Fig. 2: the authors demonstrated that pericyte-deprivation leads to an accelerated tumor progression and premature death in two glioma mouse models. Did they test if pericytes can inhibit tumor cell proliferation *in vitro*? or they exhibit this effect indirectly?

Author response, 1st revision

*Thank you for raising this concern. To address this, we have now tested the effect of pericytes on tumor cell proliferation *in vitro*. Using a transwell-based co-culture approach, we cultured Pdgf- β , p53 kd tumor spheres with pericytes or endothelial cells over 5 days. We subsequently performed a BrdU/To-Pro-3-based FACS analysis, that showed a mild proliferation-increasing effect on glioma cells for both pericytes and endothelial cells, as compared to the control group (Figure R1C). Whereas these results are in line with literature referenced in our manuscript, and underline the complex character of pericyte-tumor cell interactions, they do not question the conclusions of our work, as the model we present in our study is not based on direct interactions, but on the indirect effects of TME remodeling observed in connection with pericyte deprivation. Given the comparably small effects observed on tumor cell proliferation *in vitro*, and lack of evidence for a growth reduction of tumor cells *in vivo* in the absence of pericytes, we have decided not to include these data in the revised manuscript.*

Author response, 2nd revision

In our previous experiment presented to the Referee, we measured proliferation using BrdU incorporation. As there may be ambiguity in the interpretation of the data, we have now extended the analysis using a CFSE dilution assay. Briefly, malignant cells are labeled with the cell tracer CFSE, which is then diluted at every cell division. By flow cytometry, we analyzed the proportion of CFSE-low cells, *i.e.* cells that had undergone proliferative events. As shown in Figure 2B for reviewers, co-culturing GBM cells together with pericytes reduced the proliferative rate of the malignant cells. These data are in line with the findings from the BrdU-assay that GBM cells spend longer time in S-phase when co-cultured with pericytes, thereby reducing their relative proliferative rate. The joint conclusion from these two experiments that pericytes slightly reduce the proliferative rate of malignant GBM cells may contribute to the accelerated tumor growth observed *in vivo* in pericyte-deprived conditions. We have currently presented these data as a Figure for the Referees only, but should the Editor and Referee consider these data important to include in our manuscript, we would be happy to do so.

#10 The authors mentioned two different invasion modes of OLIG2+ tumor cells, by individual cells or by co-option with vessels (however, no evidence is provided).

Author response, 1st revision

*Using a mouse model that allows for variation in Olig2 functional status (Olig2^{cre/+}; Trp53^{fl/fl}; hEGFR^{fl/fl}), Griveau et al. analyzed vascular relationships of Olig2⁺ versus Olig2⁻ gliomas (Griveau et al., 2018, DOI [10.1016/j.ccell.2018.03.020](https://doi.org/10.1016/j.ccell.2018.03.020)), using, among other methods, IF analysis. They observed that OPC cells of Olig2⁺ tumors preferentially invade following tumor capillary tracks, *i.e.* via single-cell vessel co-option, while Olig2⁻ invading tumor cells tend to invade physiologic brain parenchyma as dense perivascular collections. Thus, the authors found different modes of invasion depending on the glioma cell state. In the mouse model used for this study, which is mechanistically different, we quantified tumor cells of the invasive zone based on IF analysis of*

Olig2⁺ GBM cells (Figures 4D, 4E and S4D), and found increased proportions of co-opting cells in the absence of pericytes. We interpreted these results as consequential to a remodeled perivascular space and possibly altered transcriptional programs of infiltrating tumor cells in pericyte-deprived gliomas, allowing for a more effective mode of glioma cell infiltration of the surrounding brain parenchyma. These findings are especially interesting in light of recently published data stating an increased glioma cell invasion in recurring IDH-wildtype gliomas, known to be pericyte-deprived, that was linked to an altered tumor microenvironment (Varn et al., 2022, DOI [10.1016/j.cell.2022.04.038](https://doi.org/10.1016/j.cell.2022.04.038)).

Author response, 2nd revision

We have now included our reasoning as a response to this query in the Discussion of the revised manuscript, see page 17.

#12 Fig. 4I: the interaction between pericytes and macrophages seems to be no difference between *Pdgfbret/+* and *Pdgfbret/ret* mice, even though *Pdgfbret/ret* mice have lower numbers of pericytes. Does the impact of pericyte depletion on macrophage polarization attribute to increased endothelial cells? The mechanism is not clear here, and functional studies are needed.

Author response, 1st revision

We thank the referee for allowing us to clarify this point. The literature clearly states pericyte functions of regulating and maintaining the endothelium in the physiologic context (Armulik et al., 2010, DOI [10.1038/nature09522](https://doi.org/10.1038/nature09522); Gaengel et al., 2009, DOI [10.1161/ATVBAHA.107.161521](https://doi.org/10.1161/ATVBAHA.107.161521)), while we and others find a diminished pericyte:endothelial cell ratio to lead to a remodeled and more defective tumor vasculature (Figure 2 A-F, (Hong et al., 2015, DOI [10.1093/jnci/djv209](https://doi.org/10.1093/jnci/djv209)) in terms of morphology, functionality and paracrine signaling. Due to the co-localization of perivascular macrophages and endothelial cells, the upregulation of several cell-cell interaction pathways that affect polarization of macrophages (e.g. CD200, GAS, CSF, TGF β), and an altered signaling of macrophages to glioma cells that we find in this work, we conclude that a remodeled endothelium shifts macrophages toward a tumor-promoting state in pericyte-poor gliomas.

To further prove this point, we have now performed an experiment in which we added the ligand GAS6, which was upregulated by the endothelium in the absence of pericytes, to cultured macrophages and evaluated its influence on macrophage polarization. We found GAS6 to reinforce alternative activation of macrophages in synergy with IL4, and have included these results now in page 9, paragraph 1 and Figure S7B in the revised manuscript.

Author response, 2nd revision

The Referee correctly points out that there are few changes in the nature of the paracrine interactions between pericytes and macrophages in pericyte-poor conditions (we would like to point out that there is still ~20% pericyte coverage left in the tumors formed in *Pdgfb^{ret/ret}* mice). Instead, we identified qualitative differences in the signaling towards macrophages emanating from the endothelial compartment. Three pathways previously identified as instigators of alternative activation of macrophages – CD200, CSF-1, and GAS6 – were exclusively found to be expressed by endothelial cells during pericyte-poor conditions. Out of these, we experimentally validated that GAS6 polarized primary macrophages towards an ‘M2’ phenotype (Supplemental Figure 8B). Notably, CD200, CSF-1 and GAS6 were not expressed by the endothelium in tumors from wildtype mice, strongly suggesting that the increased frequency of alternatively activated macrophages in tumors from *Pdgfb^{ret/ret}* mice resulted from qualitative differences, not simply due to the hyperproliferative endothelium.

We believe that these observations, demonstrating a potential novel role for the peri-vascular niche in regulating polarization of macrophages, are interesting and deserving of reporting, but have toned down previous statements in the revised manuscript.

#13 The authors demonstrated that three signaling axes known to induce macrophage polarization were exclusively active in Pdgfbret/ret endothelial cells that signaled towards macrophages. However, from their in vitro data showing that only pericyte supernatant, not endothelial cell supernatant, decreases IL4-related macrophage polarization. How to explain these inconsistent results? In addition to polarization, how about macrophage infiltration?

Author response, 1st revision

We thank the reviewer for pointing out this apparent inconsistency. It should be emphasized that the assessment of macrophage polarization by endothelial- or pericyte-conditioned medium was performed in vitro, and thus may not fully represent the more complex situation in vivo. Thus, it is not clear whether the phenotype of endothelial cells cultured on stiff plastic would be more similar to the pericyte-rich or -poor situation. Moreover, the IL4-induced alternative activation of macrophages used in the experimental setup yields an almost full polarization, making it difficult to appreciate a synergistic effect with other factors. Taken together, the main point that we would like to make with this experiment is the conceptually novel possibility that pericytes directly impinge on macrophage polarization markers, although more work is needed to mechanistically probe this function.

In our additional immune marker profiling, we found a significant increase in CD45⁺ cells in Pdgfb^{ret/ret} tumors that, considering the relative abundance of all other immune cell populations, can only arise from myeloid cells (Figure S4E). The higher abundance of macrophages and other myeloid cells in Pdgfb^{ret/ret} tumors is corroborated by our scRNA-seq analysis (Table S3) and in line with published data (Hong et al., 2015).

Author response, 2nd revision

In addition to our previous response to the Referee's query, we have softened some of the conclusions in the Results section, page 9.

#21 MET⁺ tumor cells are more aggressive in vivo. However, the in vitro data from Fig. 7E showed an opposite result. Further studies are needed to dissect the molecular mechanisms underlying this phenomenon.

Author response, 1st revision

It has been demonstrated that the MET/HGF axis can potentially transform neural stem cells into glioma-initiating stem cells, and reinforce tumor cell invasiveness (Li et al, 2011, DOI [10.1073/pnas.1016912108](https://doi.org/10.1073/pnas.1016912108); Joo 2012, DOI [10.1158/0008-5472.CAN-11-3760](https://doi.org/10.1158/0008-5472.CAN-11-3760)). Although HGF-autocrine glioma cell activation has been shown (Qin 2020, DOI [10.1093/noajnl/vdaa067](https://doi.org/10.1093/noajnl/vdaa067)), our data indicate paracrine signaling with macrophages as the primary source for HGF (Figure 7A and 7B).

In vitro experiments, lacking HGF availability from the microenvironment, hamper the MET/HGF axis, which is reflected in the lower sphere induction potential of MET⁺ compared to MET⁻ control cells that might exploit other growth-promoting pathways. In a new experiment, we have added HGF to MET⁺ and MET⁻ cell cultures in vitro to mimic the microenvironmental supply of this ligand during pericyte-poor conditions, and compared sphere induction capability. In line with our

hypothesis, we observed HGF treatment to induce a significant increase in sphere induction potential in MET⁺ as compared to MET⁻ cells. We did not find differences in cell numbers in this experiment, most likely because cell number is not always correlated with sphere induction rate, but changes become obvious during later passages. These data are now discussed in the Results section on page 13-14, and included as Figure 7E and 7F in the revised manuscript.

Author response, 2nd revision

In order to better reflect the results from our experiment (Figure 7E-F), demonstrating that the addition of HGF is necessary for efficient sphere formation from isolated primary MET⁺ GBM cells, we have rephrased parts of the Results section, pages 13 and 14, and removed all references to proliferation.

Reviewer #4 (Remarks to the Author):

Author response, 2nd revision

Thank you for contributing to valuable insights during the review process.

Figure R1

MP markers from Abdelfattah et al. 2023

A

B

C

D

E

Figure R2

A

B

Additional experimental procedures (related to figure R1C).

Glioma cells derived from PDGFB-induced, p53 kd murine tumors were grown as spheres. 2×10^4 passaged single glioma cells were cultured in transwells (0.4 μm pore size, Costar) together with 5×10^4 pericytes, endothelial cells or glioma cells (controls) for 5d. Subsequently, the samples were labeled with BrdU for 5h (Thermofisher, #8811-6600-42), stained with To-Pro-3 (ToPro) and analyzed with a BD FACS Melody system. FACS data was analyzed with FlowJo, and tested for statistical significance with t-test.

Legend to figure R1. Additional data.

(A) Expression of 9 gene lists representing myeloid cell populations taken from Abdelfattah et al. (2023) in the scRNAseq dataset.

(B) Representative fluorescence images of *Pdgfb*^{ret/+} glioma derived tissue section immune stainings for PODXL and PDGFR β , indicating the abundance of pericytes in the glioma model.

(C) Transwell assays of PDGFB, p53 kd tumor spheres, cultured with pericytes or endothelial cells over 5d, and subsequent BrdU/To-PRO-3-based FACS analysis.

(D) UMAP plot showing cell cycle states of the scRNA-seq dataset.

(E) Pericyte cluster signature score of the scRNA-seq dataset, plotted onto the spatial dataset.

* $p < 0.05$, ns not significant

Additional experimental procedures (related to figure R2B).

Co-culture experiments were performed as stated above. Proliferation analysis was conducted with a CellTrace Cell Proliferation kit (ThermoFisher SCIENTIFIC, #C34570) according to the manufacturers instructions. Cells were analyzed with a BD FACS Melody system. FACS data was analyzed with FlowJo, and tested for statistical significance with t-test.

Legend to figure R2. Additional data.

(A) Representative fluorescence images of *p53*^{-/-} *H-Ras* tumor cell- and *tva*-PDGFB virus induced glioma sections, immunostained for PDGFR β and ACTA2. Arrows indicate larger PDGFR β ⁺, ACTA2⁺ blood vessels. Arrow heads indicate smaller PDGFR β ⁺, ACTA2⁻ blood vessels.

(B) Transwell assays of PDGFB, p53 kd tumor spheres, cultured with pericytes or endothelial cells over 5d, and subsequent CSFE-based FACS analysis. Shown are the CSFE^{low} cells, representing cells that have undergone proliferation.

** $p < 0.01$, ns not significant